

# Large-eddy simulation of radiation fog: Part 1: Impact of dynamics on microphysics

Marie Mazoyer[1], Christine Lac[1], Odile Thouron[2], Thierry Bergot[1], Valery Masson[1], and
Luc Musson-Genon[3]

[1]CNRM (CNRS-Meteo-France), UMR3589, Toulouse, France
[2]CERFACS, Toulouse, France
[3]CEREA, France

*Correspondence to:* Christine Lac (christine.lac@meteo.fr)

**Abstract.**

Large Eddy Simulations (LES) of a radiation fog event occurring during ParisFog experiment have been studied with a view of analyzing the impact of the dynamics on the microphysics. The LES, performed with the Meso-NH model at 5 m resolution horizontally and 1 m vertically, and with a 2-moment microphysical scheme, included the drag effect of a trees barrier and deposition on vegetation. The model shows a good agreement with the measurements of the near surface dynamic and thermodynamic parameters as well as the cloud water content, but overestimates the cloud droplet sizes and concentration. The blocking effect of the trees induced elevated fog formation, like in the observation, and horizontal heterogeneities, and limited the cooling and the cloud water production. The deposition process was found to exert the most significant impact on the fog prediction, as it not only erodes the fog near the surface, but also modifies the fog life cycle and induces vertical heterogeneities. The comparison with the 2 m horizontal resolution simulation exhibited small differences meaning that the grid convergence was achieved. Conversely, increasing numerical diffusion through a wind advection operator of lower order led to an overestimation of the near surface microphysical fields and had almost a similar effect than removing the effect of the trees barrier. This study allows to establish the major dynamical ingredients necessary to perform correctly the fog life cycle prediction at high resolution.

## 1  Introduction

Despite the long interest in understanding fog processes, uncertainties still exist on the physical mechanisms driving fog variability. Forecasting fog remains a challenge due to the diversity of mechanisms involved during the fog life cycle and their interactions: local dynamics, turbulence, radiation, microphysics, aerosols, surface effects. Several field experiments have been carried out since the 1970's that brought important progress in fog processes understanding. Among them the campaigns from Cardington in UK (Roach et al., 1976; Price, 2011), Fog-82 in Albany (Meyer et al., 1986), Lille 91 in France (Guedalia and Bergot, 1994), Po Valley in Italy (Fuzzi et al., 1998) and ParisFog in France (Haeffelin et al., 2010). Most of them have naturally included measurement of fog droplet spectra, and set liquid water content (LWC) in the range of $0.01 - 0.4\,\mathrm{g\,m^{-3}}$ and droplet number concentration ($N_c$) of a few tens to a hundred $\mathrm{cm^{-3}}$. Hence Roach et al. (1976) related values of LWC



between 0.05 and 0.22 $\mathrm{g\,m^{-3}}$ and $N_c$ between 30 and 100 $\mathrm{cm^{-3}}$ for winter fog cases at Cardington. More recently, Mazoyer et al. (2016) reported $N_c$ for radiation fog less than 150 $\mathrm{cm^{-3}}$ over 3 winters during ParisFog.

Many important features of fog have also been characterized using one-dimensional (1D) modelling (Bergot et al. (2007), Tardif (2007), Stolaki et al. (2015) among others). But explicitly simulating turbulence motions in a 3D manner has become

necessary to improve our understanding of the physical mechanisms involved in a fog layer, since Nakanishi (2000) who was the first to use large-eddy simulation (LES). LES is a turbulence modelling technique in which most of the energy-containing eddies are explicitly resolved while eddies smaller than a certain cutoff scale, usually taken equal to the grid spacing, are parametrized by the turbulence scheme. Since then, Porson et al. (2011) explored the static stability in a fog layer, and Bergot (2013) showed the various organized structures occurring in a fog layer, which cannot be resolved in 1D. Thanks to these

studies, the dynamical characteristics of the radiation fog are more clearly identified during the three stages of the fog life cycle defined by Nakanishi (2000): the onset, the development and the dissipation phases. During the formation phase, small stripes occur in the middle of the fog layer, identified by Bergot (2013) as Kelvin-Helmotz (KH) billows, sometimes associated to a burst of turbulent kinetic energy (TKE) (Nakanishi (2000) and Bergot (2013)) but not always (Porson et al. (2011)). During the development phase, the main dynamical processes move to the top of the fog layer associated to the maximum of TKE

and horizontal rolls (Bergot, 2013). During the dissipation phase, coupled processes between the ground and the top of the fog layer explain the spatial variability of fog (Bergot (2015b)). But the link between dynamics and microphysics has not been explored specifically in these LES studies.

The quality of the LES depends on the horizontal and vertical resolutions. Hence Beare and Macvean (2004) demonstrated that simulations in stable conditions converge at 2-m horizontal resolution. Very high vertical resolution is also essential to capture

the divergence of the radiative fluxes in the first metres above the surface and therefore to produce a radiative cooling necessary to the formation of fog (Duynkerke, 1999; Tardif, 2007).

So far, most of fog LES studies consider homogeneous canopies. Only Bergot et al. (2015a) took into account the effect of surface heterogeneities as buildings on radiation fog. Other studies have considered the impact of forests on turbulence structures, like Zaïdi et al. (2013) or Dupont and Brunet (2008), but not for fog situations. In this study, we will explore a LES

of a fog case observed during ParisFog and strongly influenced by trees.

Also, very few fog LES studies are based on sophisticated 2-moment microphysical schemes, allowing to represent the aerosol impact on the radiation fog life cycle. Maalick et al. (2016) studied the effects of aerosol on the radiation fog with a LES, but in a 2D configuration that could present some limitations for the dynamical patterns of the fog layer. Additionally, most of the studies, with one or two moment microphysical schemes, fail to reproduce realistic liquid water content (LWC) as they tend

to overestimate it near the ground. For instance, Zhang et al. (2014b) simulated $N_c = 800$ $\mathrm{cm^{-3}}$ and $LWC = 0.4$ $\mathrm{g\,m^{-3}}$ and Stolaki et al. (2015) simulated $N_c = 250$ $\mathrm{cm^{-3}}$ and $LWC = 0.34$ $\mathrm{g\,m^{-3}}$ near the surface, both in 1D configuration, values that are out of the range according to Mazoyer et al. (2016) considering the same site. So one question is: is there a mechanism missing, the inclusion of which might improve the modelling of microphysical fields ? Considering deposition, the interactions with the ground surface should be an important factor as already shown by Price and Clark (2014) on measurements and von

Glasow and Bott (1999) or Zhang et al. (2014b) on 1D simulations.





The goal of this study is to better understand the physical processes dominating the fog life cycle on a complex site and impacting the microphysical fields. LES modelling at very high resolution (1 m vertically and 5 m horizontally) is used with surface heterogeneities (barrier of trees) and a 2-moment microphysical scheme. Sensitivity tests will help to understand the influence of some dynamical processes on the fog life cycle with a focus on microphysical properties. To our knowledge, this is

5 the first time that an LES study of radiation fog has been performed at such high resolution with a sophisticated microphysics taking into account the effect of heterogeneities as forests on the fog dynamics and microphysics. In a second article, the impact of aerosol activation on microphysical fields will be explored specifically allowing to characterize the contribution of the different microphysical processes.

Section 2 presents the measurement set-up and the observed case, and describes the numerical model. The reference simula-

10 tion is analyzed in Section 3, and Section 4 is devoted to sensitivity tests. Finally, some conclusions are drawn and perspectives suggested in Section 5.

## 2   Experimental design and model description

### 2.1   Measurements set-up

The selected fog event has been observed during the winter 2011-2012 of the ParisFog field campaign (Haeffelin et al., 2010)

at the Sirta (Site Instrumental de Recherche par Télédétection Atmosphérique) observatory (48.713 °N and 2.208 °E). The objective of the ParisFog campaign during three winters from 2010 to 2013 was to better understand the radiative, thermodynamic, dynamic and microphysical processes during the fog life cycle. The site where the instrument platform is installed is a semi-urban area of a complex terrain including forest, lake, meadows and shrubs next to an urban agglomeration. As shown on Figure 1a, the instrumented zone is located near a forest area. Zaïdi et al. (2013) demonstrated the impact of the trees barrier

on the observed flow when the wind flows from this side, just as in our case study. The fog case has already been studied by Stolaki et al. (2015) in the 1D configuration, and the reader should refer to this study for the description of the instrumental set-up.

At the surface, temperature and humidity sensors were located between 1 and 30 m height on an instrumented mast, with 0.2 K uncertainty for temperature and 2% for relative humidity. Wind speed was measured by two ultrasonic anemometers at

25 10 m and 30 m above ground level (agl) on the same meteorological mast. Radiative fluxes were measured on a building roof at 10 m height with $5\,\mathrm{W\,m^{-2}}$ and $4\,\mathrm{W\,m^{-2}}$ uncertainties for downward and upward fluxes respectively. Two diffusometers were operated at 3 m and 18 m to provide information on the vertical visibility with an uncertainty up to 25%. Additionally, radiosondes were launched by Météo-France twice a day in Trappes (48.7°N, 2 °E), localized 15 km to the North-West of Sirta Microphysical instrumentation was presented in detail by Mazoyer et al. (2016). A Fog-Monitor 100 (FM-100) provided

particles size distribution from 2 $\mu$m to 50 $\mu$m in diameter, while the WELAS-2000 provided particle diameter distribution between 0.96 and 10 $\mu$m according to Mazoyer et al. (2016). Aerosol particles measurements were performed by a Scanning Mobility Particle Sizer (SMPS) measuring dry aerosol diameter between 10.6 and 496 nm every 5 min, and by a CCN chamber providing the CCN number concentration at different supersaturations from 0.1 to 0.5% (Roberts and Nenes, 2005). A profiler





provided Liquid Water Path (LWP) measurements with an error up to $20\,\mathrm{g\,m^{-2}}$ according to Lohnert and Crewell (2003). We do not have measurements of dewfall and fog-droplet deposition.

## 2.2 Presentation of the observed case

### 2.2.1 Dynamics and thermodynamics

The radiative fog formed at 0200 UTC on 15 November 2011 and dissipated at the ground around 1000 UTC. Favored conditions of fog were due to a ridge at 500 hPa centred over the North Sea and anticyclonic conditions near the surface. One of the features of this event is that it is an elevated fog event, formed by a cloud layer 150 m agl and followed shortly by fog at the surface. As underlined by Stolaki et al. (2015), this characteristic is very common at Sirta since $88\%$ of the radiation fog events formed during the field experiment were also elevated, but were not classified as stratus lowering, as they were followed by formation of fog at the surface. This suggests that this property could be linked and specific to the configuration of the Sirta site.

The fog case is presented according to the three phases of the fog life cycle defined by Nakanishi (2000). Before the fog onset, between 2200 and 0200 UTC, the surface boundary layer was stable and a near-surface cooling was observed, as well as a moistening (Fig. 2). Between 0000 and 0130 UTC, the relative humidity (RH) near the ground remained nearly constant around 97%. Wind speed at 10 m height was light (around $1.8\,\mathrm{m\,s^{-1}}$) as well as TKE, with small variability (Fig. 3). At 0200 UTC, the attenuated backscatter coefficient of the lidar increased significantly at 150 m agl (not shown) revealing the formation of liquid water at this height, while RH at the surface remained at 97%. Then the cloud base height progressively subsided during about 30 min, until it reached the ground, while the near surface temperature continued to decrease by about $1\,K$. At 0230 UTC, the apparition of fog at the ground was associated to a temperature convergence in the first 30 metres, as described in Price (2011), corresponding to a neutral layer. The downwelling longwave (LWD) radiation flux increased progressively up to $325\,\mathrm{W\,m^{-2}}$ during the development of the fog layer (Fig. 4).

Then, during the fog development and mature phases, between 0200 and 0700 UTC, the near-surface layer remained quasi-neutral and temperature at the different levels remained constant. The 10 m wind speed presented a higher temporal variability than previously, as well as TKE. Around 0400 UTC, the TKE increased significantly by $0.5\,\mathrm{m^2\,s^{-2}}$ and remained then constant, maintaining a positive vertical gradient of TKE. According to Stolaki et al. (2015) and Dabas et al. (2012), the sodar indicated that the fog top height reached a maximum height of 300 m agl during its mature phase.

At the beginning of the dissipation phase, from 0700 UTC, surface temperature increased slowly (less than 0.5 K in 2 hours) and then more significantly after 0900 UTC. At 1000 UTC, the downward SW fluxes exceeded $100\,\mathrm{W\,m^{-2}}$, while near-surface temperature had increased by $1\,K$ compared to the pre-sunrise values. 30 m TKE decreased from 0800 UTC to 1000 UTC, while 10 m TKE remained approximately constant.





### 2.2.2 Microphysics

Measurements of liquid droplet microphysics near the surface indicated a sharp increase of LWC and droplet concentration ($N_c$) at the fog onset just after 0200 UTC (Fig. 5 in solid lines), up to $N_c = 53$ cm$^{-3}$ and $LWC = 0.035$ g m$^{-3}$. They correspond to a drop of the near-surface visibility from 5000 m to less than 500 m (Fig. 6a in black line). The initial elevated structure of the fog leaded to an earlier decrease of the visibility at 18 m than at 3 m agl, with a time lag of the order of 30 min. Until 0730 UTC, $LWC$ and $N_c$ decreased then slowly, while the visibility at 3 m and 18 m (not shown) remained almost constant. Between 0730 and 0800 UTC, $LWC$ and droplet concentration at 3 m decreased strongly, allowing an increase of the visibility at 3 m up to 2000 m. At 18 m agl, the visibility remained smaller than 1300 m. But the fog at the surface reformed just after, reaching $N_c = 30$ $cm^{-3}$ and $LWC = 0.024$ g m$^{-3}$, and a visibility less than 500 m, before definitively dissipating at 1000 UTC. The particle size distribution indicated 95% of the droplets with a diameter less than 20 $\mu$m, meaning that there is probably a very small impact of the coalescence process. Sampled at 3 stages of the event, it evolved during the fog life cycle and appeared consistent with the classification of Wendisch et al. (1998) (Fig. 5d). The "initial phase "(in red, at 0250 UTC) was characterized by a small droplet size, but already a second mode between 8 and 12 $\mu$m, that persisted during the 3 stages. During the mature phase (in blue, at 0500 UTC), also called "mass transfer stage ", larger droplets are numerous, up to 22 $\mu$m. During the dissipation phase (in green, at 0700 UTC), the spectrum was between the two previous ones with a reduction of the largest droplets. Hence the spectral shape remained bimodal during the fog life cycle.

The maximum of LWP measured by the profiler was reached around 0730 UTC at the beginning of the fog dissipation phase with $70$ g m$^{-2}$ (Fig. 5c). The non-zero values ($5$ g m$^{-2}$) before the fog onset are included in the error range of the measurement.

## 2.3 Model description

### 2.3.1 Presentation of the model

The non-hydrostatic anelastic research model Meso-NH (Lafore et al., 1998) (see http://mesonh.aero.obs-mip.fr) was used here in a LES configuration. The LES was based on a 3D turbulent scheme with a prognostic turbulent kinetic energy (TKE) (Cuxart et al., 2000) and a Deardorff mixing length (Deardorff, 1980).

The atmospheric model was coupled with the ISBA surface scheme (Interaction between Soil Biosphere and Atmosphere, Noilhan and Planton (1989)) through the SURFEX model (Masson et al., 2013). This scheme simulates the exchanges of energy and water between the land surface (soil, vegetation and snow) and the atmosphere above it. It uses five prognostic equations for deep temperature, deep soil water content, surface temperature, surface soil water content and water interception storage by vegetation.

In order to take into account the impact of trees on the instrumental site, we used the drag approach developed by Aumond et al. (2013) for a vegetation canopy. Indeed, Aumond et al. (2013) and Zaïdi et al. (2013) have shown the best results of the





drag approach compared to the classical roughness law to reproduce the turbulence downstream of a forest area. It consists of introducing an additional term in the momentum and TKE equations as follows:

$$\frac{\partial \alpha}{\partial t}_{DRAG} = -C_d A_f(z)\alpha\sqrt{u^2 + v^2} \tag{1}$$

where $\alpha$ represents $u$ and $v$ horizontal wind components and $TKE$, $C_d$ is the drag coefficient, set as 0.2, and $A_f(z)$ is the canopy area density, representing the surface area of the trees facing the flow per unit volume of canopy. It is a combination of the product of the fraction of vegetation in the grid cell by the leaf area index (LAI) and a weighting function that represents the shape of the trees. The vertical profile is presented in Aumond et al. (2013). We have considered atlantic coast broad leaved trees.

For the microphysics, the model included a two-moment bulk warm microphysical scheme (Khairoutdinov and Kogan, 2000; Geoffroy et al., 2008), that considers droplet concentration $N_c$ and mixing ratio $r_c$ as prognostic variables for the fog. An additional prognostic variable $N_{ccn}$ is used to account for already activated CCN, following the activation scheme of Cohard et al. (2000c). The aerosols are assumed to be lognormally distributed and the activation spectrum is prescribed as:

$$N_{ccn} = C S_{max}{}^k F(\mu, k/2, k/2 + 1, -\beta S_{max}{}^2) \tag{2}$$

where $N_{ccn}$ is the concentration of activated aerosol, $F(a, b; c; x)$ is the hypergeometric function, $C\ (m^{-3})$ is the concentration of aerosols, and $k$,$\mu$ and $\beta$ are adjustable shape parameters associated with the characteristics of the aerosol size spectrum such as the geometric mean radius ($\bar{r}$) and the geometric standard deviation ($\sigma$), as well as solubility of the aerosols ($\varepsilon_m$) and temperature ($T$) (see below the values for our case study). $S_{max}$ is the maximum of supersaturation, verifying $\frac{dS}{dt} = 0$. The evolution of $S$ includes three terms account respectively the effects of a convective ascent using vertical velocity $w$, the growth of droplets by condensation for the new activated droplets, and a radiative cooling, as in Zhang et al. (2014b):

$$\frac{dS}{dt} = \phi_1 w - \phi_2 \frac{dr_c}{dt} + \phi_3 \frac{dT}{dt}|_{RAD} \tag{3}$$

where $\phi_1(T)$, $\phi_2(T,P)$ and $\phi_3(T)$ are functions of temperature and pressure. Following Pruppacher et al. (1998) and after simplification, $S_{max}$ can be diagnosed by:

$$S_{max}{}^{k+2}.F(\mu, k/2, k/2 + 1, -\beta S_{max}{}^2) = \frac{(\phi_1 w + \phi_3 \frac{dT}{dt}|_{RAD})^{3/2}}{2kc\pi\rho_w \phi_2{}^{3/2} B(k/2, 3/2)} \tag{4}$$

with $B$ the Beta function and $\rho_w$ the density of water.

The condensation/evaporation rate is derived using the Langlois (1973) saturation adjustment scheme. The cloud droplet sedimentation is computed by considering a Stokes law for the cloud droplet sedimentation velocity and by assuming that the cloud droplet size distribution $n_c(D)$ fit a generalized Gamma law:

$$n_c(D) = N_c \frac{\alpha}{\Gamma(\nu)} \lambda^{\alpha\nu} D^{\alpha\nu - 1} exp(-(\lambda D)^\alpha) \tag{5}$$



where $\lambda$ is the slope parameter, depending on the prognostic variables $r_c$ and $N_c$:

$$\lambda = (\frac{\pi}{6}\rho_w \frac{\Gamma(\nu + 3/\alpha)}{\Gamma(\nu)} \frac{N_c}{\rho_a r_c})^{1/3} \qquad (6)$$

$\alpha$ and $\nu$ are the parameters of the Gamma law. They were adjusted using droplet spectra measurements from the FM-100 database of our case study and were set at $\alpha = 1$ and $\nu = 8$. These parameters are also used for the radiative transfer.

In addition to droplet sedimentation, fog deposition is also introduced which represents direct droplet interception by the plant canopies. In nature, it results from turbulent exchange of fog water between the air and the surface underneath leading to collection (Lovett et al., 1997). Here, the fog deposition flux $F_{DEP}$ is predicted at the first level of the atmospheric model (50 cm height) for grass area, and over the 15 m height for trees, in a simplistic way following Zhang et al. (2014b): $F_{DEP} = \rho_a \chi V_{DEP}$ with $\chi = r_c, N_c$ where $V_{DEP}$ is the deposition velocity. In a review based on measurements and parametriza-

tions, Katata (2014) showed that $V_{DEP}$ values ranged from 2.1 to $8.0\,\mathrm{cm\,s^{-1}}$ for short vegetation. A more complete approach would be to include a dependance of $V_{DEP}$ with momentum transport and also with LAI, but we supposed here that $V_{DEP}$ is constant, equal to $2\,\mathrm{cm\,s^{-1}}$. A sensitivity test to this value will be presented. Water sedimentation and deposition amounts are supplied to the humidity storage of the surface model.

The radiative transfer was computed with the ECMWF radiation code, using the Rapid Radiation Transfer Model (RRTM,

Mlawer et al. (1997)) for longwave and Morcrette (1991) for shortwave radiations. Cloud optical properties for LW and SW radiation took into account cloud droplet concentration in addition to cloud mixing ratio.

### 2.3.2   Diagnostics of visibility

Visibility can be diagnosed assuming an exponential scattering law:

$$VIS = -\frac{ln\varepsilon}{\beta} \qquad (7)$$

with $\beta$ the extinction coefficient, and using a visual range defined by a liminal contrast $\varepsilon$ of 0.02 (Koschmeider, 1924). The most common parametrizations used to diagnose the visibility with droplet properties in models with 1-moment microphysical schemes are expressed as:

$$VIS = \frac{a}{LWC^b} \qquad (8)$$

where $a$ is 0.027 and $b$ is 0.88 for Kunkel (1984) (units of LWC and VIS are $\mathrm{g\,m^{-3}}$ and $km$ resp.).

When droplet concencentration $N_c$ is taken into account with 2-moment microphysical schemes, the diagnostic becomes:

$$VIS = \frac{c}{(LWC.N_c)^d} \qquad (9)$$

where $c$ is 1.002 and $d$ is 0.6473 for Gultepe et al. (2006) developed with eastern Canada observations, and $c$ is 0.187 and $d$ is 0.34 for Zhang et al. (2014a) from polluted North China Plain measurements.

Measurements of visibility can be employed to estimate the validity of visibility diagnostics used for models. Hence, the three

formulations were applied to the observed $LWC$ and $N_c$ and compared to the observed visibility in order to determine which





one fits the best the observed values (Fig. 6a). Zhang et al. (2014a) parametrization was the most adapted to the observations of our case study, as it is more sensitive to low $LWC$ and $N_c$ values, even if it tended to underestimate slightly the observed visibility. Diagnostics from Kunkel (1984), and Gultepe et al. (2006) even more, underestimated the 3 m observed visibility in our case study.

### 2.3.3 Simulation set-up

For the reference simulation (noted REF), the horizontal resolution was 5 m over a domain size of 200 x 200 grid points. 126 vertical levels were used between the soil and the top of the model at 1500 m. The vertical resolution was 1 m for the first 50 m and increased then slightly above. Momentum variables were transported with a fourth-order centred scheme (noted CEN4TH), whereas scalar variables were transported with the PPM (Piecewise Parabolic Method) scheme (Colella and Woodward, 1984). The time step was 0.1 s. The domain of simulation is presented on Figure 1b, with a trees barrier of 15 m height and 100 m wide perpendicular to the wind direction. The rest of the domain was composed of grass. The lateral boundary conditions were cyclic. The radiation scheme was called every second.

The simulation began at 2320 UTC on 14 November 2011 before the fog formation, and covered 12 h. Temperature, humidity and wind speed vertical profiles were initialized with data from the radiosonde launched in Trappes. Meteorological conditions at Trappes can differ slightly from the Sirta site. Therefore wind, temperature and humidity were modified in the nocturnal boundary layer up to 400 m agl to adjust with the data recorded at the 30 m meteorological mast at the Sirta site, as illustrated on Fig. A.1. The soil temperature and moistening were given by the soil measurements, corresponding to a surface temperature of 276 K and a soil moisture of 70%. Following the profiles from soundings, a geostrophic wind of $8\,m\,s^{-1}$ was prescribed as a forcing, without any other forcing. To generate turbulence in addition to the effect of trees, a white noise of $0.5$ K was applied in the first 100 m.

It was also necessary to characterize the aerosol size spectrum for Eq.2. The supersaturations reached in fog were lower than 0.1% meaning that the CCNC measurements were not directly usable, as shown by Hammer et al. (2014); Mazoyer et al. (2016). But when using the Kappa-Köhler theory and the SMPS observations, the aerosols concentration at supersaturations under 0.1% can be retrieved knowing the aerosol hygroscopicity ($\kappa$) at these supersaturations. This method, proposed by Mazoyer et al. (2016), has been applied to our case study the hour before the fog onset. The activation spectrum was thus computed from observation above 0.1% supersaturation, and from computation under 0.1%. A fit of this computed activation spectrum was applied according to Eq.2 (Fig. A.2a), corresponding to the aerosols particles distribution ($C = 2017$ cm$^{-3}$, $\sigma = 0.424$, $\bar{r} = 0,1, \varepsilon_m = 1$) in red on Fig. A.2b. This does not match the measured distribution (in black) nor the lognormal fitted on the accumulation mode (in blue), due to the fact that Cohard et al. (2000c) formulation has not been developed for fog low supersaturation. Nevertheless, considering that the activation spectrum was deduced from measurements, it includes a good degree of confidence.

The reference simulation will be now presented.





## 3 The reference simulation

The performance of the REF simulation is first examined, based on a comparison with observed values of thermohygrometric, dynamic, radiative and microphysical parameters near the ground. Considering that the REF simulation reached a good degree of confidence, vertical evolution and horizontal variability of the simulated fog are then characterized during the different

phases of the fog life cycle. It should be emphasized that observations localized at one point were compared to averaged simulated fields over an horizontal area located downstream the trees barrier (blue contour area of Fig. 1b) representative of the instrumental area, as we will see that there were significant horizontal heterogeneities over this area.

### 3.1 Parameters near the surface

#### 3.1.1 Dynamics and thermodynamics

Figure 2 shows the time series of near surface observed and simulated temperature and RH. At the initialization of the simulation, near surface temperature were in agreement with the observations while RH were very slightly underestimated. During the cooling before the fog onset, the model developed a too stable layer, especially in the 5 first metres between 0000 and 0100 UTC. The convergence of temperature was simulated with 30-40 min of delay compared to the observations

Considering RH near the surface, the fog started to appear around 0230 UTC. Between 0430 and 0830 UTC, simulated and

observed temperature were in fairly good agreement, with a quasi-neutral near surface layer. The fog started to dissipate from the ground at 0830 UTC, with approximately one hour and a half ahead of the local observation. This time lag induced a slight overestimation of near surface temperature, increasing up to $0.5$ K at 1100 UTC. But the negative temperature gradient near the surface representative of the development of the convective boundary layer was quite well reproduced after the beginning of the dissipation.

Dynamical fields at 10 m and 30 m were fairly well reproduced by the model (Fig. 3 in red): the 10 m wind speed was in good agreement with the observation during all the simulation. Until 0300 UTC, a quasi linear increase of TKE was produced by the model with a TKE at 10 m agl higher than at 30 m contrary to the observations (Fig. 3). Around 0300 UTC, a more sudden increase occurred like in the observations, even if it was underestimated. Then simulated TKE remained quasi constant from 0400 UTC around $0.7 \, \mathrm{m^2 \, s^{-2}}$, with a higher variability than before. The model developed similar TKE values at 10 m and 30

25   m, while 30 m observed values were higher.

Considering the radiative fluxes (Fig. 4), the increase of the LWD flux associated to fog onset was simulated with a delay of 30-40 minutes, meaning that there was a delay on the formation of fog at elevated levels. After that, the LWD flux of $325 \, \mathrm{W \, m^{-2}}$ was correctly reproduced, indicating that the temperature and the optical thickness of the fog were fairly well simulated. Observations developed a difference of $8 \, \mathrm{W \, m^{-2}}$ between LWU and LWD during the fog life cycle, but the model

failed to reproduce this difference, leading to a slight underestimation of LWU. If measurements did not encounter an error, this probably means that the radiative properties of the simulated surface were not perfectly represented. A test on the emissivity of the surface (1 instead of 0.96) had no impact on the radiative fluxes, suggesting that the soil temperature was probably





underestimated. After sunrise (0659 UTC), the downward and upward SW fluxes were overestimated by $15 \, \mathrm{W \, m^{-2}}$, and LWD were underestimated in a similar way due to the advanced dissipation time.

### 3.1.2 Microphysics

Considering the microphysical fields at 3 m agl, the onset of LWC higher than $0.001 \, \mathrm{g \, m^{-3}}$ was in agreement with the obser-
vations (Fig. 5b). The delay on LWD flux increase was not reproduced on LWC, meaning that the time of formation of fog at the ground was correctly reproduced but the previous formation at elevated levels was underestimated. This is corroborated by the LWP evolution (Fig. 5c), also characterized by a 40 min delay compared to the Sirta ponctual observation.

The increase of LWC during the development phase was in agreement with the observed one but this phase was too long leading to an overestimation, with a maximum value of $0.07 \, \mathrm{g \, m^{-3}}$ instead of $0.035 \, \mathrm{g \, m^{-3}}$ observed. Then, during the mature phase, the slow decrease of LWC was reproduced, up to 0830 where both observed and simulated values became less than $0.001 \, \mathrm{g \, m^{-3}}$. But as we have seen before, in reality, this first event of fog dissipation only concerned the very near surface levels, as observed visibility at 18 m remained less than 1300 m. On the contrary, the fog did not reformed near the surface in the simulation, inducing an advance on the dissipation time. The discrepancies between simulation and observation was higher on cloud droplet concentration than on LWC during all the fog life cycle, as the model strongly overestimated $N_c$, up to a factor of 7 (maximum values of $350 \, \mathrm{cm^{-3}}$ simulated against $53 \, \mathrm{cm^{-3}}$ observed, Fig. 5a). Maxima of $N_c$ and LWC occurred at the same time, around 0300 UTC, than both decreased. But $N_c$ increased again during the dissipation phase, before dropping sharply at the end of the fog.

The droplet size distribution (DSD) in the model is described by the normalized form of the generalized gamma distribution which gives a monomodal form (Fig. 5d). During the formation phase (red lines), the model overestimated small droplets with a diameter between 2.5 $\mu$m and 7.5 $\mu$m and did not produce droplets of diameter larger than 9 $\mu$m. This trend continued during the fog life cycle (blue and then green line) even if it was less marked than at the initial stage. The model produced the largest droplets at the mature stage like in the observations, before reducing the spectrum during the dissipation. The simulated modes corresponded to 4 $\mu$m, 7.5 $\mu$m and 6 $\mu$m of diameters at the 3 stages. The overestimation of small droplets and the underestimation of larger ones leaded to the weakness of droplet sedimentation. Indeed, the surface cloud water amount by sedimentation is negligible after 12 hours of simulation (around $10^{-4}$mm), while it reached 0.0674 mm by deposition. The weakness of droplet sedimentation could partly explain the overestimation of $N_c$ during all the fog life cycle, as well as the LWC, as it kept too much water in the fog layer. But another reason that could explain the overestimation of droplet concentration and that will be developed in the Part 2 of this study is that the equation (3) allowing to compute the supersaturation peak value does not take into account the sink term due to pre-existing LWC, as explained in Thouron et al. (2012).

Due to the overestimation of simulated droplet concentrations, the Zhang diagnostic of visibility applied to simulated microphysical fields underestimated the observed visibility at 3 m and 18 m (Fig. 6). The Gultepe formulation is better adapted to our simulation, reproducing correctly the visibility drop at the onset of the fog, while the visibility remained slightly underestimated during the fog life cycle. As LWC values are better reproduced than $N_c$, the Kunkel formulation matched the observations the best.





The comparison between REF simulation and observation for the set of parameters shows a fairly good agreement, even if there were some discrepancies. The main discrepancies were, considering the fog life cycle, an underestimation of the effect of elevated fog formation and an advance of 1.5 h on the dissipation time. These elements are probably partly due to the semi-idealized representation of the Sirta surface in the simulation, and also to the comparisons with ponctual observations, knowing the horizontal variability as we will see further. Considering the microphysical fields, the main discrepancy was an overestimation of small droplets concentration near the ground, and to a less degree of LWC. They are felt to be acceptable and we can therefore consider that the REF simulation can be used to explore the processes driving the fog life cycle.

## 3.2 Vertical evolution

First the fog vertical evolution is analyzed. Figure 7 represents time evolutions of vertical profiles of $r_c$ and $N_c$, radiative cooling rate and vertical velocity in the updrafts, while Figure 8acd represents the same time evolution for total turbulent kinetic energy (resolved plus subgrid, noted TKE), dynamical and thermal production of TKE for the REF simulation, all averaged over the horizontal area downstream the trees barrier. As a preliminary comment, subgrid kinetic energy is one order less than resolved kinetic energy (not shown), meaning that the 5 m horizontal resolution allows a LES approach as most of the eddies are resolved.

The evolution of $r_c$ allows to decompose formally the fog life cycle into the three phases: the formation, between 0200 and 0320 UTC, until the fog became optically thick; the development, between 0320 and 0720 UTC, until $r_c$ at upper levels of the fog layer began to decrease, and the dissipation from 0720 UTC.

Before the fog onset and during the formation phase, the TKE was small and spread over a 30 m layer that deepened slowly, consecutively to the flow induced by the trees barrier. TKE was mainly produced by dynamical production which presented maxima at two levels, near the surface and at 15 m height due to the trees. Thermal production was negative due to the thermal stratification. The radiative cooling near the ground (Fig. 7c) and the mixing by the tree drag effect were the ingredients allowing the apparition of fog at elevated level (Fig. 7a). Then the mixing by the trees barrier caused a subsiding effect of the fog layer down to the ground and a vertical development above (Fig. 7a). Hence, the effect of elevated formation was reproduced, even if the height of fog onset was underestimated (150 m given by the ceilometer and 30 m in the simulation). The period of subsiding effect of the fog to reach the ground was therefore shorter and equal to 20 min. During this first phase, mean updraft vertical velocities were small, up to $0.15 \mathrm{~m\,s}^{-1}$ (not shown), in agreement with Ye et al. (2015), who observed vertical velocity of $0.1 - 0.2 \mathrm{~m\,s}^{-1}$ in fog layer between 40 m and 220 m depth in China. Considering Eq.3 for supersaturation evolution with the two source terms function of vertical velocity and radiative cooling, activation of fog droplets was during the formation fog mainly produced by radiative cooling at the top of the fog layer (Fig. 7b and c).

At the beginning of the development phase (around 0320 UTC), when the fog reached approximately 80 m, it became optically thick to longwave radiation. Exactly at that time, TKE increased significantly by dynamical production (Fig. 8a and c), in agreement with Nakanishi (2000), meaning a dynamical change. The optical thickness of the fog layer caused a strong radiative cooling at the top of the fog layer, higher than $5.5 \mathrm{~K\,h}^{-1}$ (in absolute value, Fig. 7d), and LWC values became stronger in the upper part of the fog layer. Hence, the fog top became the location of the dominant processes with radiative





cooling. It induced small downdrafts and buoyancy reversal. Additionally to the vertical velocity of the updrafts now higher than $0.2\,\mathrm{m\,s^{-1}}$ in all the fog layer, a second maximum of droplet concentration of $1000\,\mathrm{cm^{-3}}$ occurred in the upper part of the fog layer around 0320 UTC. The sudden optical thickening corresponded to the increase of surface LWD up to $320\,\mathrm{W\,m^{-2}}$ (Fig. 4) and to the maximum of cooling at the ground. In the same time, temperatures converged between the vertical levels

near the ground (Fig. 2a and b), showing the effect of fog on the stability profile as analyzed by Price (2011).

Then, during the development phase, the top of the fog layer was characterized by vertical wind shear inducing a positive dynamical production of TKE, while small values of positive thermal production appeared at the top due to buoyancy reversal. Inside the fog layer, in the 40 lowest metres, the drag effect of the trees induced higher values of kinetic energy higher than $0.6\,\mathrm{m^2 s^2}$.

The maximum of $r_c$ continued to increase in the upper part of the fog layer up to 0500 UTC, reaching $0.35\,\mathrm{g\,kg^{-1}}$ at 120 m (Fig. 7a). In the same time, LWD surface fluxes remained constant while the fog layer continued to deepen and the LWP to increase up to 0500 UTC (Fig. 7c).

Around 0430-0500 UTC, a change occurred on the development of the fog layer: it continued to thicken but at a smaller rate, while the LWP began to decrease in the simulation. This change of growth at the top of the fog layer was associated

to a warming in the fog layer (not shown) and a decrease of the maximum radiative cooling near the top that spread over a broader depth (Fig. 7c). This corresponded also to an increase of resolved updraughts and downdraughts near the top (Fig. 7d). Variability of the fog depth became also stronger, linked to fog-top waves as we will see further. This change of growth seems to be linked to the fact that the fog layer reached the top of the nocturnal boundary layer, meeting stronger temperature, humidity and wind gradients. This increased the top entrainment process, limiting the deepening of the fog layer. With the

decrease of the top radiative cooling, cloud droplet concencentration became quasi homogeneous in the fog layer, except near the ground where it decreased by deposition. In the same way, cloud mixing ratio began to decrease also near the ground (Fig. 7b).

The beginning of the dissipation phase (around 0720 UTC) can be identified by the beginning of solar radiation, and divergence between surface LWU, starting increasing, and surface LWD, starting decreasing in the simulation (Fig. 4). The

25 dissipation of the fog began at the surface, and the fog lifted into a stratus layer. The radiative heating of the surface increased the convective structure of the fog as vertical velocity in the updrafts increased (Fig. 7b) and thermal production of TKE became significantly positive (Fig. 8d). Additionally, after sunset, downdraughts at the top of the fog layer increased solar radiation reaching the ground and feeding the heating at the base of the fog layer. Hence, near the ground, both thermal and dynamical effects contributed to the production of TKE, and to deepen the TKE layer up to 60 m. The height of the fog top

continued to increase as it was driven by radiative and evaporative cooling inducing vertical motions and top entrainment. If mixing ratio decreased at all levels, droplet concentration increased sharply when the fog layer lifted from the surface (Fig. 7b). As the cloud evolved into a stratus layer, droplet activation was no more induced by radiative cooling at the top of the fog layer but by updraft vertical velocity in all the cloud depth, and especially near the stratus base. The stronger vertical velocity (Fig. 7d) allowed to activate more droplets for the same water content amount. Droplets became smaller and more numerous,





preventing the droplet sedimentation process and limiting the decrease of LWP, while the deposition process was not active any more without cloud droplet at the surface. We will now consider horizontal heterogeneity of the fog layer.

### 3.3 Horizontal variability

To better characterize turbulent structures and the impact of trees on the fog layer, the horizontal variability of the fog layer is examined. Figure 9 presents horizontal and vertical cross-sections of wind speed, cloud mixing ratio, potential temperature and TKE at 0240 UTC during the formation phase. The trees barrier induced a blocking effect of the flow upstream, and enhanced the turbulence by wind shear downstream, accelerating the flow near the ground and creating longitudinal structures in the direction of the wind. Ascents occurred upstream and small subsidence downstream, up to $2\,\mathrm{cm\,s^{-1}}$ (not shown), drawning warmer and dryer air from above to the ground. Therefore structures of stronger wind near the ground downstream coincided with structures of warmer and clear air as they delayed the fog formation. The fog formed at the surface upstream from the trees, and 500 m downstream, while it appeared first at elevated levels between both (Fig. 9d). The fog took about 1 hour to cover the entire domain at ground level. Thus, heterogeneity of the surface vegetation explains heterogeneities on fog onset over the Sirta site, as well as the property to develop fog first at elevated levels. After the formation phase, the base of the fog layer standed at the ground over the whole domain. These results are in agreement with the building effects on fog studied by Bergot (2015b) who found a 1.5 hour period of heterogeneity of fog formation over the airport area.

During the development phase, as shown on the vertical cross-sections of Fig. 10 at 0620 UTC, horizontal rolls appeared at the top of the fog layer and were associated to dynamical production of TKE by shear. They were aligned almost perpendicular to the mean wind direction (not shown). These structures correspond to Kelvin-Helmotz (KH) instability, already observed by Uematsu et al. (2005) and modelled by Nakanishi (2000) and Bergot (2013). They had depth corresponding to about one third of the fog layer height, like in Bergot (2013), and horizontal wavelength of the order of 500 m. These horizontal rolls explain oscillations at the top of the fog layer visible on Fig. 7 and Fig. 8. They became well marked from 0430 UTC when the depth of the fog layer began to increase more slowly, as the fog layer reached the top of the nocturnal boundary layer, meeting stronger wind gradients (not shown). They induced strong horizontal variability of cloud mixing ratio near the top of the fog, with larger values in the ridge of the fog-top rolls, and smaller ones in the troughs. Local updraughts occurred upstream the crest of the wave, and downdraughts downstream, both up to $1.2\,\mathrm{m\,s^{-1}}$ (Fig. 10d). Maximum of droplet concencentration occurred near the top of the fog layer (Fig. 10b) in the radiative cooling layer (Fig. 10c), and preferentially upstream the crest of the wave than downstream, in ascent area, where they were preferentially activated and transported. These extrema of droplet concentration do not appear on Fig. 7 as they were hidden by the spatio-temporal average.

Inside the fog layer, the radiative cooling was negligible while vertical velocity presented strong spatial heterogeneities. Maxima of supersaturation appeared strongly correlated with vertical velocity (Fig. 10e), with values up to $0.27\%$ which were probably overestimated even if measurements of supersaturation peaks were not available beyond the surface. But droplet concencentration was quasi homogeneous over the horizontal domain, and did not show a strong correlation to the maximum supersaturation, due to the pre-existing droplets.





Near the ground, maximum simulated values lay around $0.1\%$ while Hammer et al. (2014); Mazoyer et al. (2016) reported observed supersaturation peaks lower than $0.1\%$. The presence of trees and the deposition process induced smaller droplet mixing ratio and concencentration near the surface.

During the dissipation phase, heterogeneities remain at the top of the fog layer, but the signature of KH waves disappeared (not shown). The dissipation of fog at ground level took about 20 minutes, and did not reveal a clear effect of surface heterogeneity, as well as in Bergot et al. (2015a).

Having characterized vertical and horizontal heterogeneities of the fog during its life cycle, sensitivity tests are now presented to identify the sources of variability and their impact on the microphysical fields.

## 4 Sensitivity study

In order to better characterize the physical processes dominating the fog life cycle and driving the microphysical properties, sensitivity tests are conducted in a second step. The resulting simulations are summarized in Tab.1 considering their difference with the REF simulation.

### 4.1 Impact of trees

To evaluate the impact of trees on the dynamics and on the microphysics of the fog, a simulation called NTR has been run, where the grass has replaced the barrier of trees. Hence, deposition on the grass is considered over the whole domain. Without trees, 10 m wind speed was overestimated over the instrumental area (Fig. 3a). As in REF, the model developed a sudden increase of TKE at 0300 UTC at the beginning of the development phase, meaning that this change was linked to the increase of the optical thickness, and not to the turbulence induced by trees (Fig. 3b and Fig. 8b). But after this period, TKE was underestimated and remained stronger at 10 m height than at 30 m, contrary to observation, which means that the drag effect of trees was responsible of the observed stronger TKE at 30 m height. The fact that the REF simulation developed quasi similar TKE at 10 m and 30 m agl probably means that the representation of surface heterogeneities was still underestimated, which can be explained by the broad range of surface covers in reality in addition to the trees (lake, small buildings ...), not included in the simulation.

The main differences on dynamics between NTR and REF appeared first on total TKE, with the absence of stronger values in the first 40 metres in NTR, as they were restricted to the immediate vicinity of the ground (Fig. 8b).

Before the formation, the too thin layer of turbulence near the ground in NTR limited the supply of warmer air from above, inducing an overestimation of the vertical temperature gradient before the fog, and emphasizing the cooling in the low levels, with 2 K less than in REF (Fig. 2c). Figure 11a presents the temporal evolution of cloud mixing ratio vertical profiles during the NTR simulation, to be compared to Fig. 7a and b for REF, and Figure 12a and b exhibited instantaneous vertical cross section of potential temperature at the fog formation with REF and NTR. The stronger cooling with NTR homogeneized the fog formation at the ground and prevented elevated fog formation. The consequence is that the onset of fog with NTR occurred 2 hours before the observation and the REF simulation (Fig. 2d). Fig. 14 summarizes the impact of sensitivity tests on the





microphysical fields and NTR (purple lines) can be compared to REF (red lines) in Fig. 14abc. During the formation and the development phases, the depth of the fog layer was thinner in NTR than in REF, because of the formation at the ground and the absence of mixing without trees, thus limiting the vertical development. Maximum of cloud mixing ratio with NTR was increased compared to REF, due to the absence of warming by entrainment, leading to a cooling largely overestimated near

the ground when compared to observations (Fig. 14a). Therefore the Kunkel diagnostic underestimated the visibility, as well as the other diagnostics (Fig. 6d). Inside the fog layer, despite the increase of $r_c$, the production of $N_c$ was not higher than in REF (Fig. 11b), as smaller vertical velocities compensated the stronger cooling in the activation process.

Additionnally, near the ground, droplet concentration was even smaller than in REF, as deposition effect, acting only at the first vertical level in NTR, was active since the onset of the fog, due to the absence of elevated formation and to the thinner

fog layer. Consequently, the DSD at 3 m shifted towards larger droplets in NTR (Fig. 14c), consistently with the reduction of droplet concentration.

Also, during the development phase, 500 m wavelengths of KH waves were more smooth and regular without trees (Fig. 13) and this has been noted during all the phase. This can be shown on kinetic energy spectra applied on vertical velocity over the whole fog depth, computed according to Ricard et al. (2013) and presented on Fig. 16. The spectra of REF and NTR

presented two main differences: firstly the TKE variance was smaller with NTR at wavelengths finer than 200 m, meaning that the flow presented less fine scale structures without the tree drag effect. Secondly, the peak of variance at 500 m wavelength, corresponding to the KH waves, was more pronounced with NTR.

The regular KH waves with NTR induced a regular wave pattern of the radiative cooling layer at the top of the fog layer (Fig. 13c). Therefore, higher droplet concentrations were spread over a deeper layer at the top of the fog with NTR than with

REF (Fig. 13b). This is also emphasized by the fact that the pre-existing cloud water content, higher with NTR than with REF, is not taken into account in the diagnostic of maximum supersaturation as it should be. Comparing Fig. 13 to Fig. 10, it also appears that vertical velocity associated to KH waves at the top of the fog were smaller with NTR than with REF, but this was not systematic during the period. However, the intensity of vertical velocity at the top of the fog layer seems to be correlated to the depth of the KH waves. Hence, it appears that surface heterogeneities relative to the trees introduced small perturbations

up to the top of the fog layer on this case, that modified the regular wave pattern but that did not remove the KH waves.

During the dissipation phase, KH waves at the top of the fog layer remained longer in NTR as the dissipation time was delayed (not shown). This time lag was in better agreement with the observations, unlike the rest of the fog life cycle

To summarize, the absence of trees barrier produced an unrealistic simulation, as it induced a too early onset of fog (2 hours of advance), a too strong cooling in the low levels, and a large overestimation of the near surface LWC during all the fog

life cycle, damaging the visibility. On the other side, droplet activation was reduced near the ground due to smaller vertical velocities and to a stronger impact of surface deposition, shifting the DSD to larger droplets. It also modified the signature of the KH waves at the top of the fog layer, with a more regular pattern and less heterogeneities on the microphysical fields near the top of the fog layer. The impact of the deposition process will now be examined more precisely.





## 4.2 Impact of deposition

Two simulations have been carried out to better characterize the role of the deposition process, both keeping the trees barrier. The first one, called NDT, removed only deposition over trees compared to REF, considering that trees act as grass for deposition. This was done by activating deposition only at the first level of the model. The second one, called NDG, removed

fully deposition. Figure 14abc compares near surface microphysical fields. NDT increased slightly droplet mass and number downstream the trees barrier, as well as the LWP during the fog life cycle (Fig. 15). But removing deposition everywhere with NDG had a considerable impact as it increased by a ratio of 8 the cloud mixing ratio and the concentration near the surface.

With NDG, the onset of fog occurred at the surface and not at elevated levels, almost 2 hours before observation and REF simulation (Fig. 11c). During the development phase, vertical gradient of $r_c$ and $N_c$ have disappeared (Fig. 11c and d), even

if radiative cooling at the top was stronger with higher cloud mixing ratio (with maxima of cooling more than $-8 \, \mathrm{K \, h^{-1}}$). The temporal evolution of cloud droplet concentration in the fog layer shows constant vertical profiles, without maxima during the formation and the dissipation phases like in REF. Hence, cloud droplet concentration was constant during the fog life cycle near the ground, while observations reported a decrease during the development phase (Fig. 14a). NDG developed also a broader DSD, with more numerous larger droplets. Therefore, droplet sedimentation was significantly increased as NDG

reported a mean cumulated cloud water amount of 0.053 mm reaching the surface during the 12 hours by sedimentation, while the REF simulation produced 0.067 mm of cloud water at the surface after 12 hours by deposition, the sedimented water being negligible.

The fog layer was also deeper during all the life cycle, and therefore the LWP was largely overestimated with a maximum between 0500 and 0600 UTC, about twice the observed value (Fig. 15). Due to the larger amount of cloud water, the dissipation

at the ground was delayed.

Another test, noted DE5, considered a deposition velocity $V_{DEP}$ of $5 \, \mathrm{cm \, s^{-1}}$ instead of $2 \, \mathrm{cm \, s^{-1}}$ like in REF (Fig. 14abc and Fig. 15a). It induced a slight diminution of the near surface LWC, but the fog life cycle, the droplet concentration and the LWP remained almost unchanged, meaning that the deposition process is not too highly sensitive to the deposition velocity.

Zhang et al. (2014b) had already shown that taking into account a deposition term in simulations seemed to have some effect

on the droplet concentration in the layer near the ground and consequently on visibility. But their effect was less pronounced than here. A possible explanation is that both $u*$, the friction velocity, and the mean volumetric diameter of droplets, taken into account in their parametrization, were underestimated. In our case, the deposition process with a simple parametrization appeared to be essential to correctly simulate the fog life cycle and to be closer to the observed microphysical values near the ground. It impacted significantly the microphysical fields. Hence, the remove of this process induced droplet sedimentation,

but in insufficient quantity to avoid unrealistic droplet concentration and mixing ratio in the fog layer and near the surface. It also modified the fog life cycle in terms of onset and dissipation times, LWP and microphysical characteristics inside the fog layer, and prevented elevated fog formation which was a climatological characteristic of the Sirta site. We will now examine the impact of the horizontal resolution to the simulated fog life cycle.





### 4.3 Sensitivity to effective resolution

In order to assess the impact of spatial resolution on the fog life cycle, a 2 m horizontal resolution (called DX2) was carried out using the same momentum advection scheme than REF (CEN4TH). According to Skamarock (2004), kinetic energy (KE) spectra deduced from simulations allow to set up the effective resolution as the scale from which the model departs from the

5 theoretical slope, which is $-3$ for vertical velocity spectra applied to stable turbulence. Mean KE spectra applied to vertical wind component revealed effective resolution of the order of $4 - 5\ \Delta x$ for simulations with CEN4TH (DX2 and REF), in agreement with Ricard et al. (2013), namely 8 m and 20 m respectively (Fig. 16).

With DX2, top entrainment was more active as updrafts and downdrafts were represented at finer resolution, limiting the cooling near the surface (Fig. 12d) and the vertical development of the fog. The LWP was therefore slightly reduced (Fig.

15b) but the fog onset and dissipation times and the LWC were almost unchanged (Fig. 14e). Only small droplets were more numerous, increasing slightly droplet concentration during all the fog life cycle (Fig. 14d and f). The close results between DX2 and REF are in agreement with the convergence in stable conditions around 2 m resolution already shown by Beare and Macvean (2004).

Then two other tests have been held on the wind transport scheme, keeping the 5 m horizontal resolution: the CEN4TH

scheme has been replaced by the WENO (Weighted Non-Oscillatory, Shu (1998)) scheme at 3rd order (called WE3) or 5th order (called WE5). These spatial schemes, associated to Explicit Runge-Kutta temporal scheme, allow time steps 10 times larger than CEN4TH associated to Leap-Frog temporal scheme, but they were run here with the same small time step (0.1 s) for the comparison. Due to the upstream spatial discretization, WENO schemes were implicitly diffusive and were therefore characterized by a coarser effective resolution, especially WENO3 due to its lower order: the effective resolutions were 35 m

(i.e. $7\ \Delta x$) and 70 m (i.e. $14\ \Delta x$) for WE5 and WE3 respectively (Fig. 16).

WE3 reduced significantly the top entrainment and the supply of warmer and dryer air from above, emphasizing the cooling near the surface (Fig. 12c). Indeed, the diffusive contribution of the advection operator dissipated small updrafts and suppressed a part of the resolved kinetic energy variance, in particular the one present at the top of the fog layer. This induced an overestimation of the thermal gradient near the surface before the fog, and a too strong cooling by 1 K during the fog (not shown).

The consequences of the enhanced cooling were that the onset of fog at the surface happened 1.5 h before observation (with a preformation at elevated levels that is not shown), the LWC during all the fog life cycle was largely overestimated, and the dissipation time was delayed (Fig. 14e). The DSD moved towards larger droplets (Fig. 14.f). It increased the droplet sedimentation as the mean cloud water content reaching the surface by sedimentation was $4.10^{-4}$ mm after 12 hours of simulations, that is 4 times more than in REF, compared to $0.1$ mm by deposition for WE3. Considering the microphysical fields, WE3

tends to be closer to NTR simulation, meaning that a diffusive transport scheme dilutes significantly the tree drag effect.

On the contrary, the differences were very small between WE5 and REF: only the LWP was a bit higher with WE5 during the dissipation phase due to a fog slightly deeper. This underlines the less diffusive behaviour of WENO5 and its higher accuracy compared to WENO3.





Thus the jump on the effective resolution with the diffusive WENO3 scheme affected significantly the fog life cycle, while the smaller deviation with WENO5 had almost no impact. Increasing numerical implicit diffusion seemed almost similar than removing the drag effect of trees. This has also underlined the importance of the numerical schemes in order to correctly handle the cloud edge problem (Baba and Takahashi, 2013). As well a 2 m horizontal resolution instead of 5 m did not bring important

changes.

Finally, sensitivity tests on initial fields are presented.

### 4.4 Sensitivity to initial conditions

A test has been held on initial humidity field in order to see if it reduced the bias on microphysical fields. Two simulations were considered, the first one, called HM2, where the relative humidity of the initial profile in the boundary layer has been reduced

by $2\%$, and the second one, called HP3, where the relative humidity of the initial profile has been increased by $3\%$ over the same depth. It appeared that the fog life cycle was significantly modified, with a fog onset time around 2 hours before and after with HP3 and HM2 respectively, deviating from the observations (Fig. 14g, h and i). But surface LWC almost coincided during the development and mature phases. Also both simulations do not change the DSD, as well as the droplet concentration extrema. LWP of REF, HM2 and HP3 were superimposed (Fig. 15c) during the mature phase, therefore the dissipation time

was unchanged.

It appears that taking away some humidity in the initial state would not allow to reduce the droplet concentration, and the overestimation of the droplet concentration could not be explained by an inadequate initial humidity profile.

Sensitivity tests have also been conducted on surface temperature $(+-\ 2\ \text{K})$ and humidity $(+-\ 10\%)$, but had very small effects on the fog life cycle and on the microphysical fields (not shown).

The last test involved an increase (VP3) or a decrease (VM3) of the wind speed in the free atmosphere in the initial and forcing conditions. The lower the wind, the earlier the formation time, the higher the LWC and the later the dissipation time, as the mixing with upper dry and warm air is reduced (Fig. 14g, h and i). On the contrary a stronger wind reduced drastically the duration of the fog life cycle and the surface LWC. VM3 succeeded to broaden the droplet spectrum, but the extrema of the droplet concentration did not change significatively.

Therefore, all the tests presented on Figures 14 and 15 failed to reduce the droplet concentration compared to REF. Only the NTR simulation reduced it somewhat due to a broader droplet spectrum, but it overestimated the LWC and diverged on the fog life cycle in the same way. This probably means that the way to improve the droplet concentration prediction does not lie in the dynamical conditions any more, considering the improvement brought by the deposition process.

### 5 Conclusion

Large eddy simulations of a radiation fog event observed during ParisFog campaign were performed, with the aim of studying the impact of dynamics on microphysics. In order to study the local structures of the fog depth, simulations were performed at 5 m resolution on the horizontal and 1 m on the vertical near the ground, and included a trees barrier present near the





instrumental site, taken into account in the model with a drag approach. The model included a 2-moment microphysical scheme, and a deposition term was added to the droplet sedimentation, representing the droplets interception by the plant canopies and acting only at the first vertical level above grass, and over the height of the trees.

The performance of the reference simulation was satisfactory as there was a fairly good agreement with the classical near-
surface measurements. The main discrepancy was an overestimation of small droplets concentration near the ground, to a less degree of liquid water content, and an advance on the dissipation time of little more than one hour. The good performance allowed to explore the processes driving the fog life cycle.

The formation of the fog at elevated levels and the rapid subsiding effect of the fog layer down to the ground just after, that is a frequently observed characteristic of radiation fog events at the Sirta site, has been elucidated as a consequence of the tree
drag effect as the wind overcame this obstacle. In contrast, the fog formed at the surface upstream from the trees and 500 m downstream, leading to about one hour of duration for the formation at the surface over the whole domain.

At the begin of the development phase, the fog became optically thick to longwave radiation, inducing a significant increase of kinetic energy by dynamical production, and that was also associated to temperature convergence at low levels. The radiative cooling near the top of the fog layer was the main source of droplet activation so that the droplet concentration was maximum
in the upper levels of the cloud.

During the development phase, a slower growth of the fog layer depth occurred when the fog layer reached the top of the nocturnal boundary layer, meeting stronger thermodynamical and dynamical gradients. Horizontal rolls at the top of the fog layer, associated to Kelvin-Helmotz instabilities, became well-marked. The cloud droplet concentration became quasi homogeneous in the fog layer on time average. But locally, extrema of droplet concentration occurred near the top of the fog in the
radiative cooling layer, with maxima preferentially upstream the crest of the waves than downstream, in ascent area, meaning that mainly vertical velocity and secondly radiative cooling contribute to droplet activation at the top of the fog layer. Inside the cloud layer, maxima of supersaturation were directly linked to the local updrafts, while the droplet concencentration remained almost homogeneous.

During the dissipation phase, as the fog involved into a stratus layer, cloud mixing ratio decreased at all levels but a sharp
increase of the droplet concentration occurred over the whole depth of the cloud as droplets were now only activated by the convective ascents at the base of the stratus.

Then different sensitivity tests provide a better understanding of the physical processes involved during the fog life cycle. The absence of the trees barrier produced an unrealistic fog simulation, with a too early onset, a too strong cooling and a large overestimation of the surface LWC, damaging the visibility diagnostic. It also modified the signature of the KH waves at the
top of the fog layer, with a more regular pattern shown on energy spectra.

The removal of the deposition process over all the vegetation canopy exerted the most significant impact on the fog prediction, as it produced unrealistic water content near the surface, prevented elevated fog formation, but also modified the fog life cycle and suppressed vertical and temporal heterogeneities of the microphysical fields.

Increasing the horizontal resolution up to 2 m did not change significantly the fog prediction, meaning that a grid convergence
seems to be achieved at these resolutions. Conversely, increasing the numerical diffusion with a momentum transport scheme





of lower order, which involves a coarser effective resolution, limited drastically the top entrainment, and tended almost to the solution where the tree drag effect was ignored, underlying the importance of the properties of numerical schemes in LES, in particular at cloud edges.

Endly, modifying the initial conditions in terms of humidity or wind profiles impacted the fog life cycle but failed to reduce much more the number concentration.

This study demonstrates the feasibility and the interest of LES including surface heterogeneities to improve our understanding of the fog processes. At these fine resolutions, surface heterogeneities have a strong impact which explains a part of the variability in the fog layer. These simulations remain very challenging. Therefore, horizontal and vertical variabilities of fog layer need also to be much more explored in future field experiments. The horizontal variability especially at the onset of the fog also underlines that a point observation may not be very representative for what happens in a coarser grid box of a numerical weather prediction model for instance.

One of the main point of this study is that fog water deposition cannot be neglected anymore in 3D fog forecast models, as still often occurs. It not only influences microphysical fields near the ground but also the whole fog life cycle. It seems to be more important than droplet sedimentation in our case, keeping in mind that the concentration of small droplets was overestimated. In this study, the deposition term has been introduced crudely and this would need some refinements in further studies. It would need to be proportional to the wind speed, and it could also consider the hygroscopic nature of canopies. By analogy with dry deposition, it would also be better to take into account droplet diameter, supposing that this field is correctly reproduced. Other studies have also shown that fog water deposition was strongly enhanced at the forest edge, up to 1.5-4 times larger than that in closed forest canopies (Katata, 2014), so it could be interesting to simulate the edge effect of fog water deposition. It is also crucial to perform measurements of fog water deposition and dewfal during field experiments (Price and Clark, 2014).

This study has shown the strong importance of some dynamical effects which operate at a first order to predict correctly the fog life cycle. But among all the tests carried out, no one has succeeded to reproduce correctly the droplet concentration, always overestimated. Now that the fog life cycle is correctly reproduced on this case, trying to correct this defect appears as the priority. Thouron et al. (2012) have developed a new scheme based on a supersaturation prognostic variable to avoid excessive droplet concentration in 2-moment microphysical schemes, as they demonstrated that some assumptions of the adjustment process are not valid anymore with LES. One of the main points is to take into account the pre-existing cloud water as a sink of supersaturation, in order to limit the activation of cloud droplets. The relevance of this scheme, applied in Thouron et al. (2012) to cumulus and stratocumulus clouds, needs to be demonstrated for fog clouds, and this will be the subject of the second part of this study.

*Acknowledgements.* Authors are very grateful to all SIRTA operators and database managers. This research was partially funded by the European Community's Seventh Framework Program (FP7/2007-2013) under the SESAR WP 11.2.2 project, under Grant Agreement 11-120809-C




| Name of the simulation | Difference of configuration with REF |
| --- | --- |
| NTR | No TRee: homogene surface |
| NDT | No Deposition on Trees |
| NDG | No Deposition (on Grass and trees) |
| DE5 | Deposition velocity equal to $5\ \mathrm{cm\,s^{-1}}$ |
| DX2 | Horizontal resolution = 2m |
| WE3 | 3rd order WENO advection for momentum |
| WE5 | 5th order WENO advection for momentum |
| HM2 | Initial RH minus $2\%$ |
| HP3 | Initial RH plus $3\%$ |
| VM3 | Geostrophic wind minus $3\ \mathrm{m\,s^{-1}}$ |
| VP3 | Geostrophic wind plus $3\ \mathrm{m\,s^{-1}}$ |

**Table 1.** Simulation configurations for sentivity tests

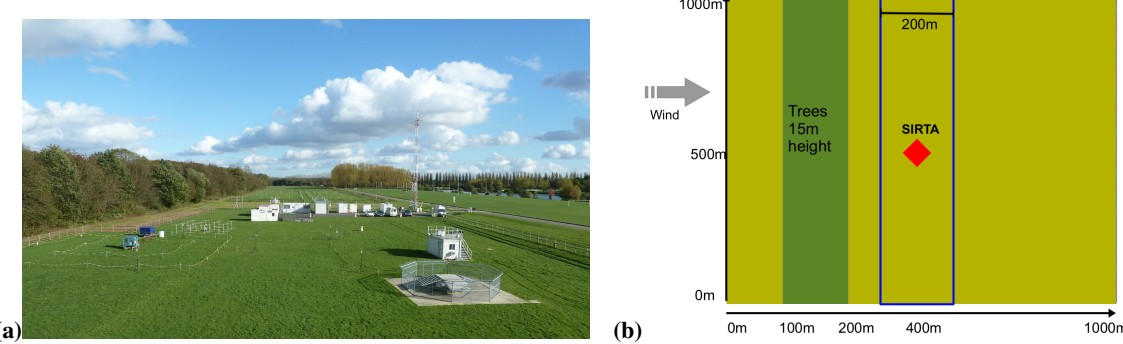

**Figure 1.** View of the measurement site (a) and modelling domain (b) with the trees barrier: all the simulated averaged results are presented on the blue contour area.



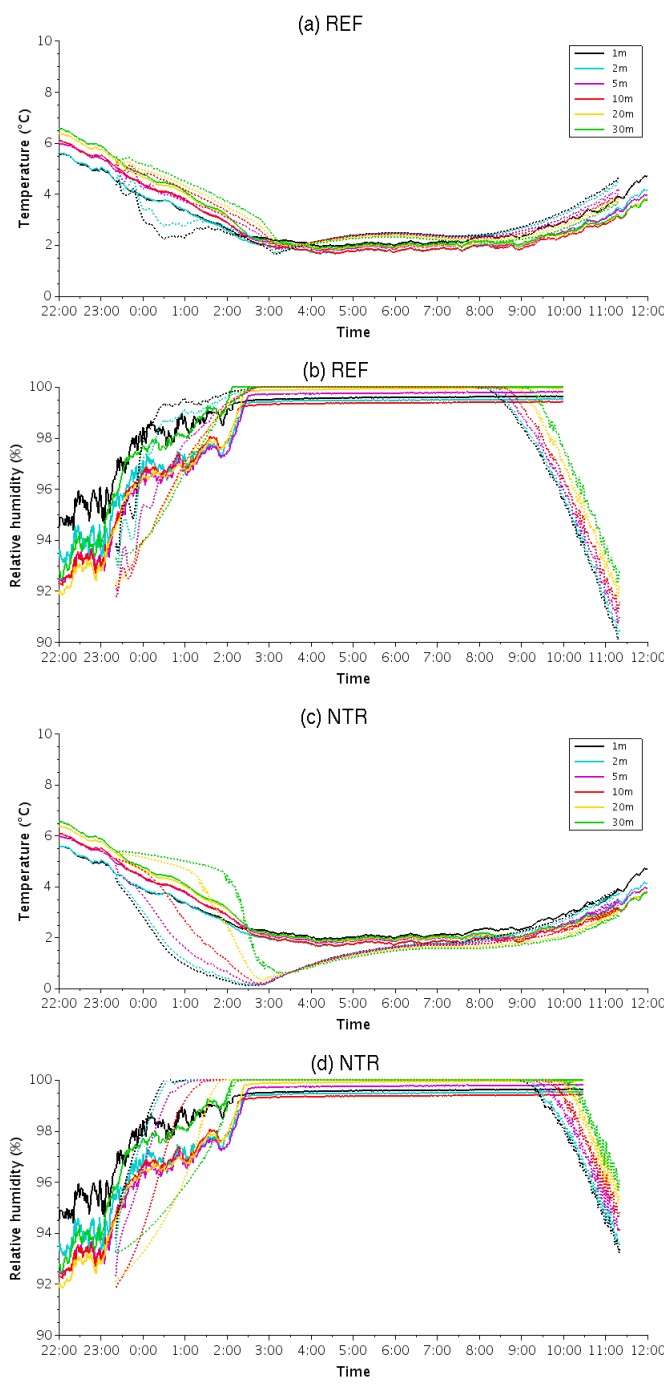

**Figure 2.** Observed (solid lines) and simulated (dashed lines) temporal evolution of temperature (a and c) and relative humidity (b and d) at 1m, 2m, 5m, 10m, 20m and 30m for the REF (a and b) and the NTR (without trees) (c and d) simulations.





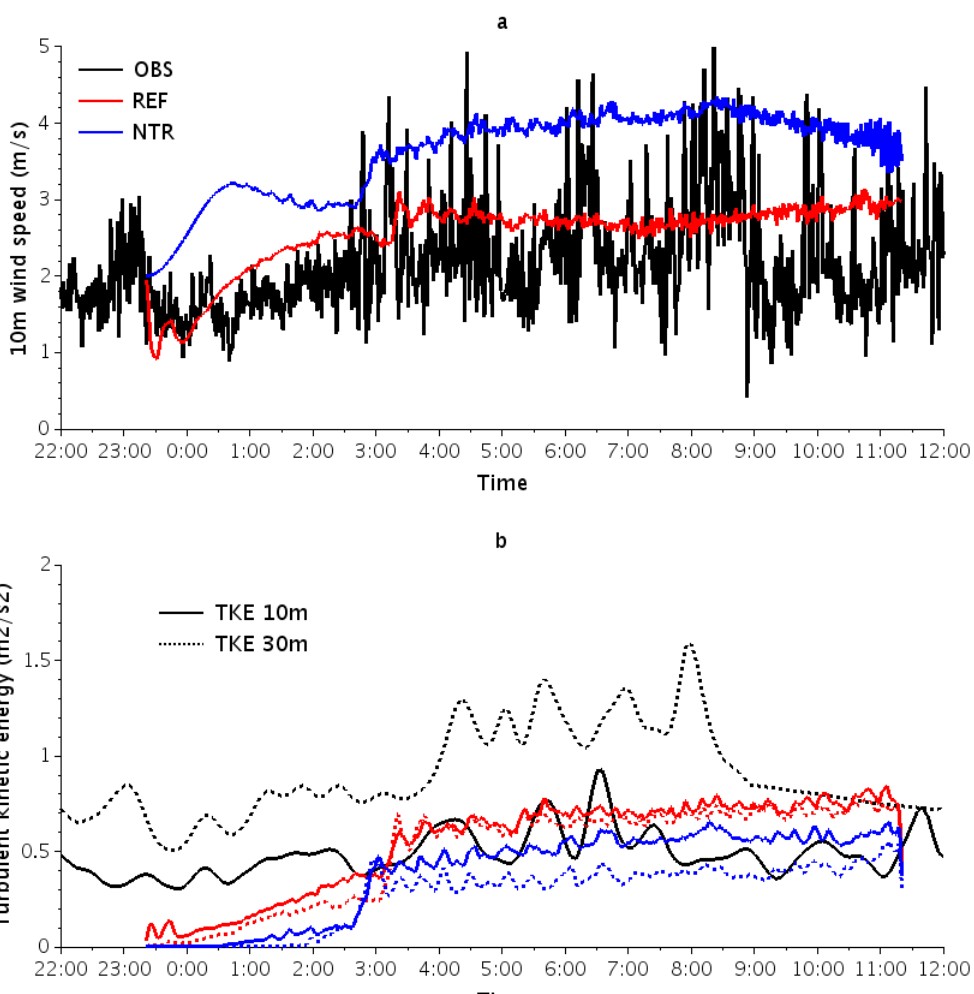

**Figure 3.** Observed (black lines) and simulated (color lines) temporal evolution of 10m wind speed (a) and 10m (solid line) and 30m (dotted line) TKE (b) for the REF (red line) and the NTR (without trees) (blue line) simulations.



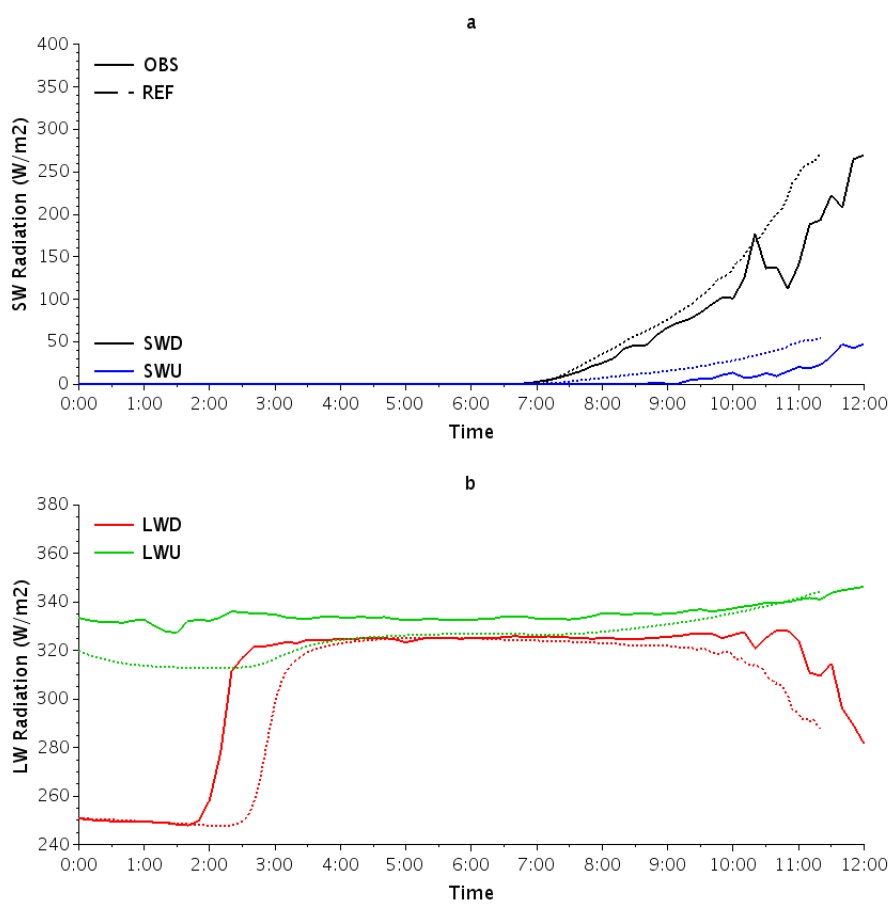

**Figure 4.** Observed (solid lines) and simulated (dotted lines, with the REF simulation) temporal evolution of downward and upward (at 1m) shortwave (a) and longwave (b) radiation fluxes (in $W/m^2$).





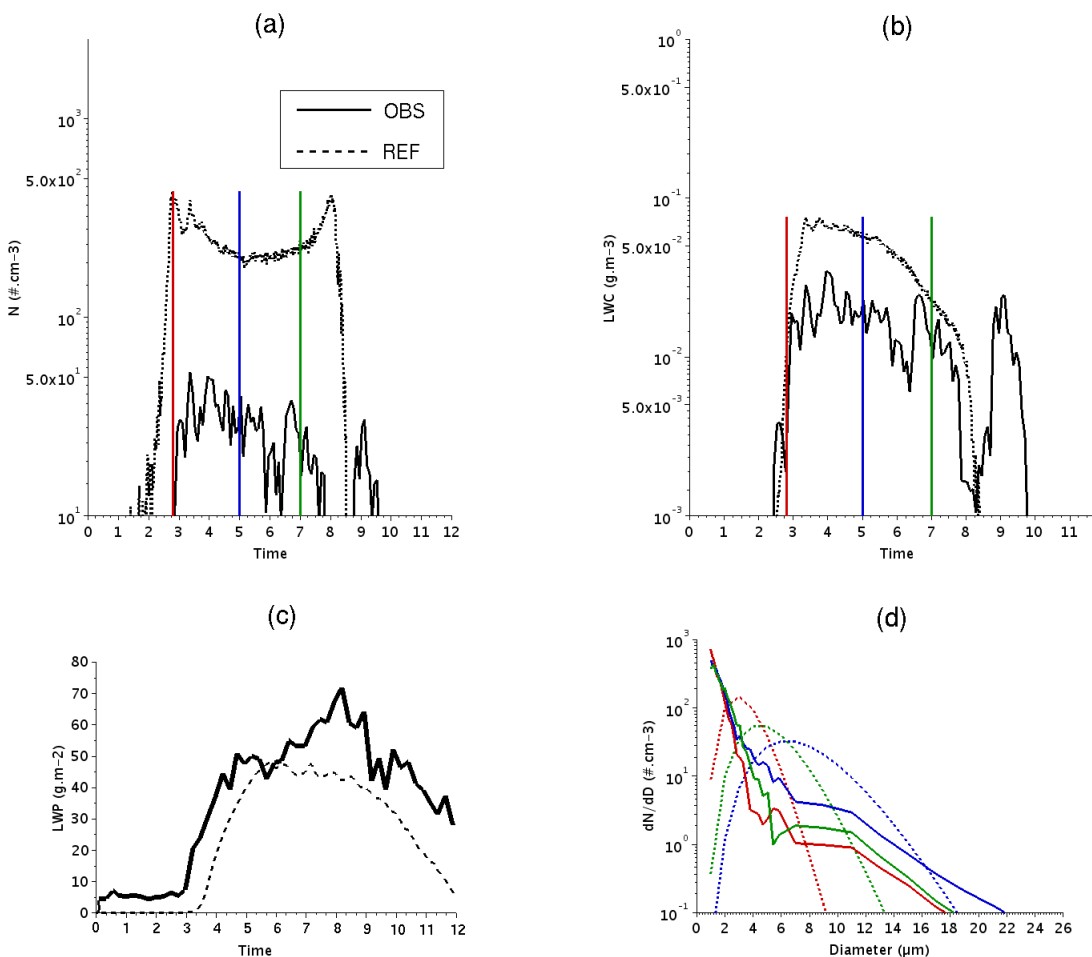

**Figure 5.** Time series of droplet concentration (a, in $\mathrm{cm}^{-3}$), liquid water content (b, in $\mathrm{g\,m}^{3}$), and LWP (c, in $\mathrm{g\,m}^{-2}$), and particle size distribution (d, in $\mathrm{cm}^{-3}$) at 0250 UTC (in red), 0500 UTC (in blue) and 0700 UTC (in green) at 3 m agl observed (in solid line), and simulated by REF (in dotted line).





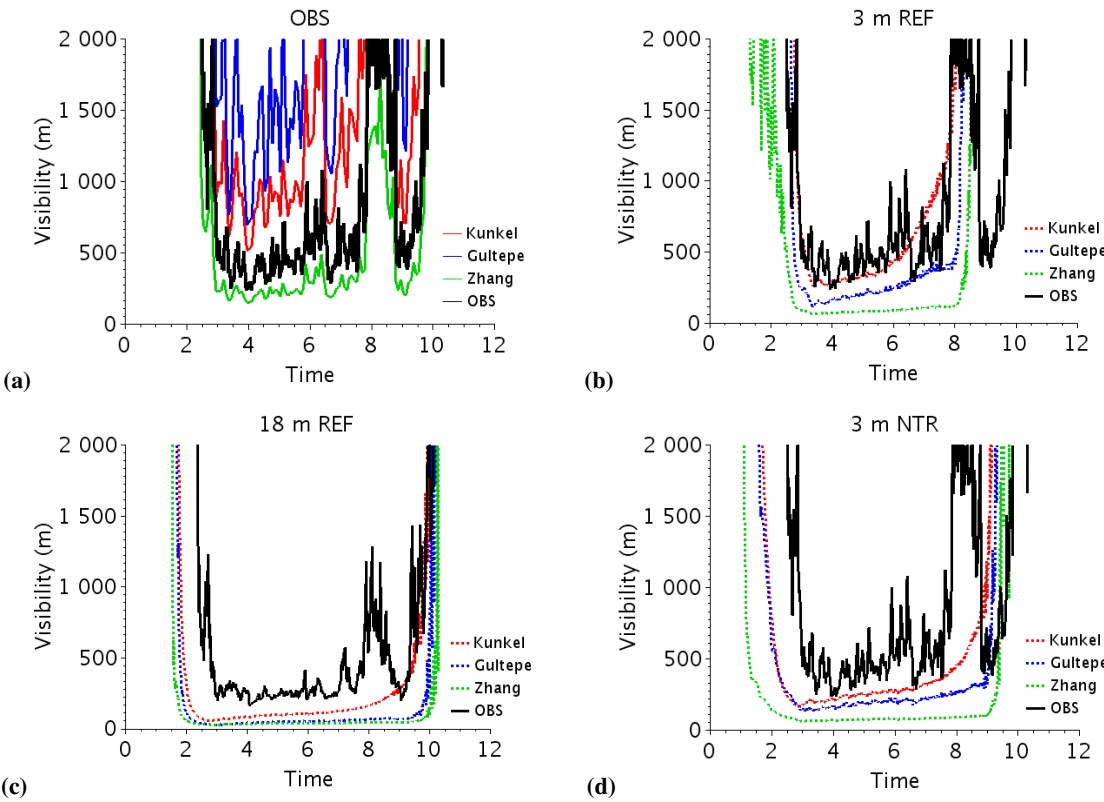

**Figure 6.** (a) 3 m observed (in black) and diagnosed (in colour) visibility with the observed microphysical fields according to Kunkel (1984), Gultepe et al. (2006) and Zhang et al. (2014a) (in m). (b) and (c) 3 m and 18m visibility diagnosed with the microphysical fields from the REF simulation. (d) 3 m visibility diagnosed with the microphysical fields from the NTR simulation (in m).





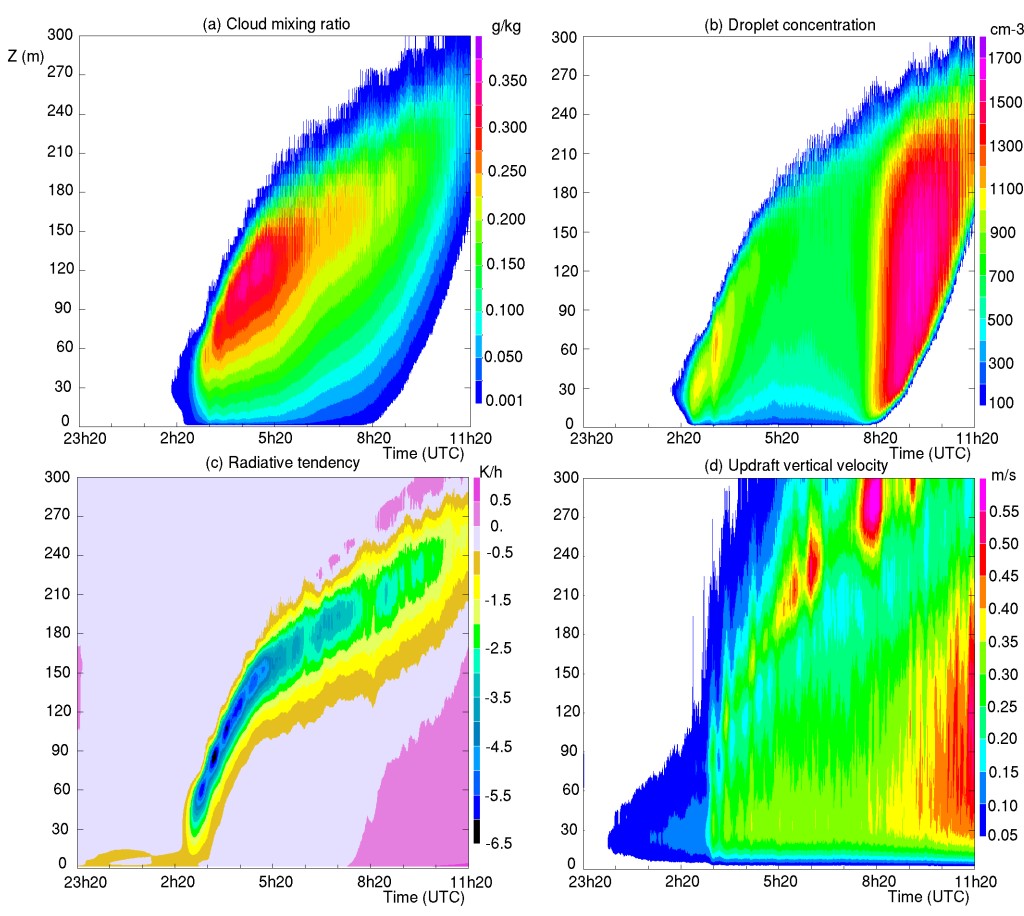

**Figure 7.** Temporal evolution of simulated vertical profiles of cloud mixing ratio (a, in $\mathrm{g\,kg^{-1}}$), droplet concentration (b, in $\mathrm{cm^{-3}}$), radiative tendency (c, in $\mathrm{K/h}$) and updraft vertical velocity (d, in $\mathrm{m\,s^{-1}}$) for the REF simulation.



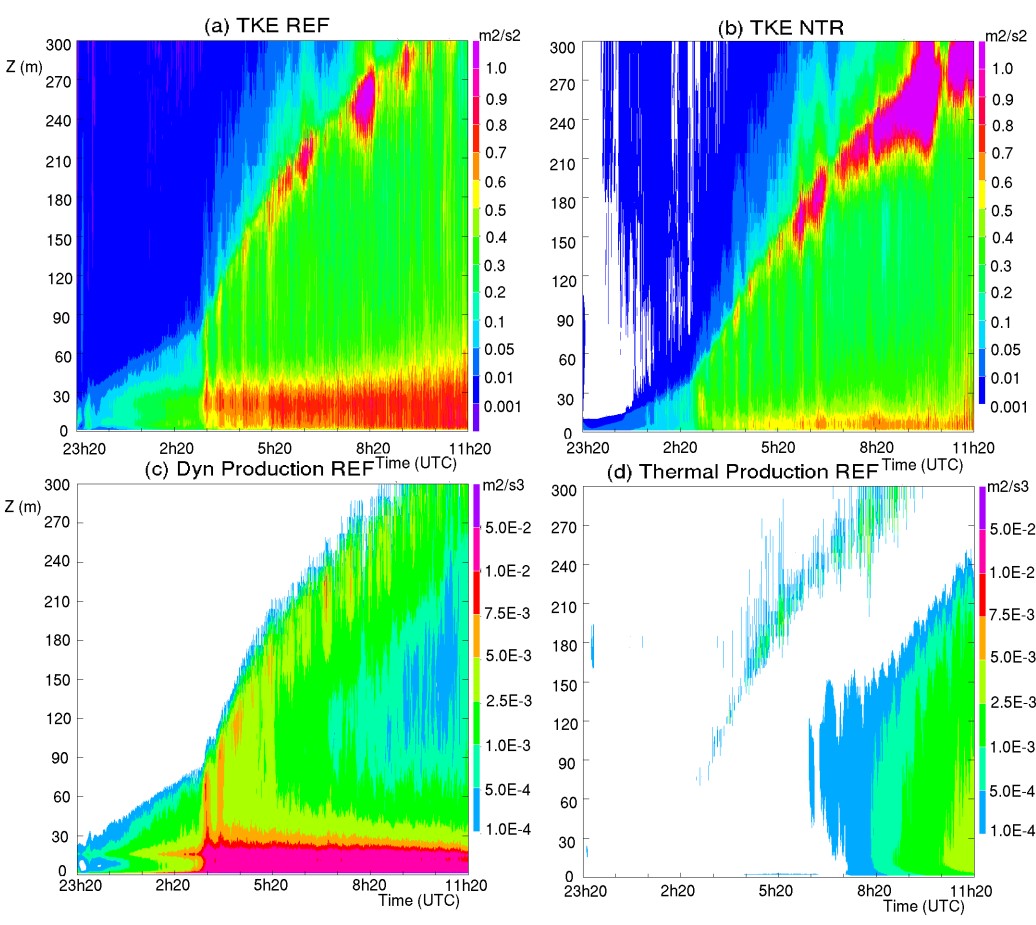

**Figure 8.** Temporal evolution of mean vertical profiles of total (resolved+subgrid) turbulent kinetic energy (in $m^2\,s^{-2}$) for REF (a) and NTR (b) simulations, and dynamical (c) and thermal (d) production of total turbulent kinetic energy (in $m^2.s^{-3}$) for the REF simulation.





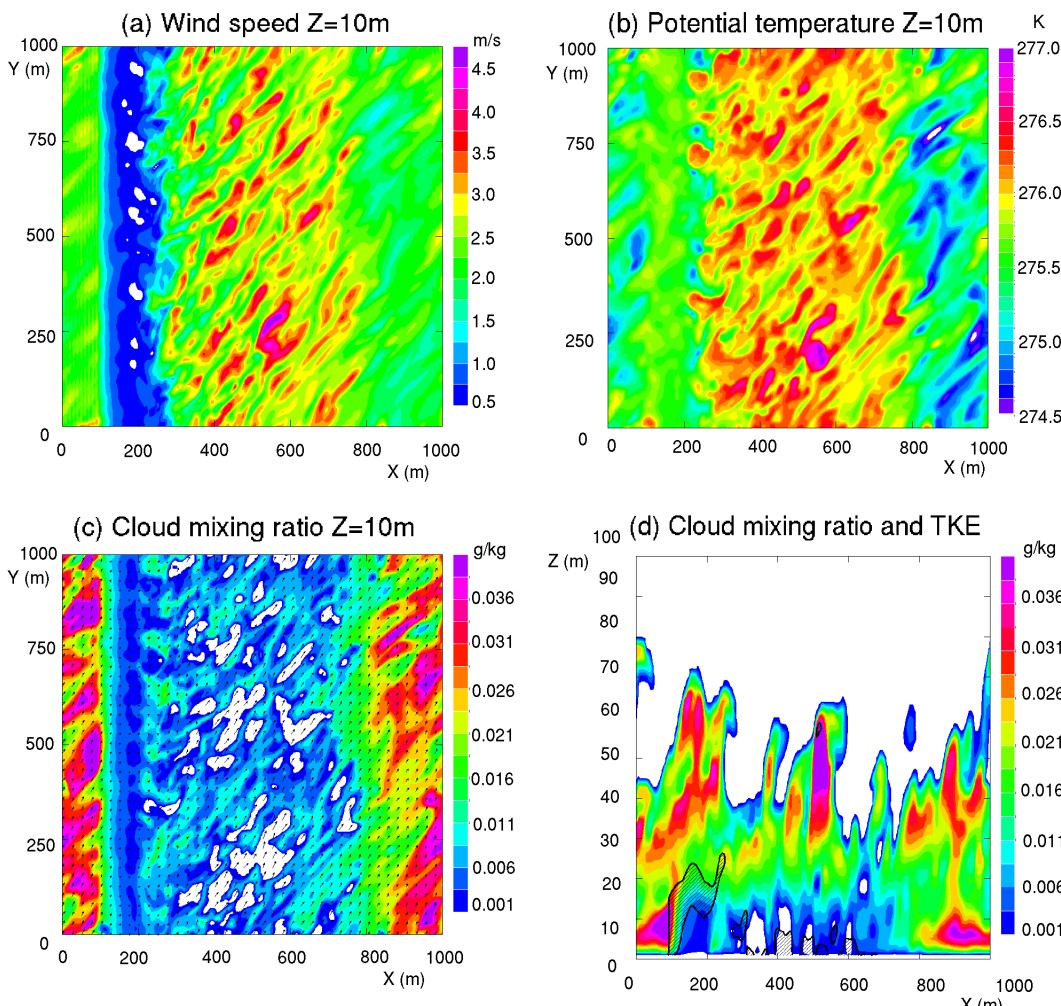

**Figure 9.** REF simulation at 0240 UTC: (a), (b) and (c): Horizontal cross-section at 10 m height of wind speed (a, in $\mathrm{m\,s^{-1}}$), cloud mixing ratio (b, in $\mathrm{g\,kg^{-1}}$) and potential temperature (c, in K). (d): Vertical cross-section at Y=500m of cloud mixing ratio (in $\mathrm{g\,kg^{-1}}$) with area of TKE higher than $0.1\ \mathrm{m^2\,s^{-2}}$ shaded.





**Figure 10.** Vertical cross-section at Y=500m at 0620 UTC for the REF simulation: (a) Cloud mixing ratio (in $\mathrm{g\,kg^{-1}}$), (b) Droplet concencentration (in $\mathrm{cm^{-3}}$), (c) radiative tendency ( in K/h, (d) vertical velocity (in $\mathrm{m\,s^{-1}}$) and (e) maximum of supersaturation (in %) with the isoline of $r_c = 0.01\,\mathrm{g\,kg^{-1}}$ superimposed.



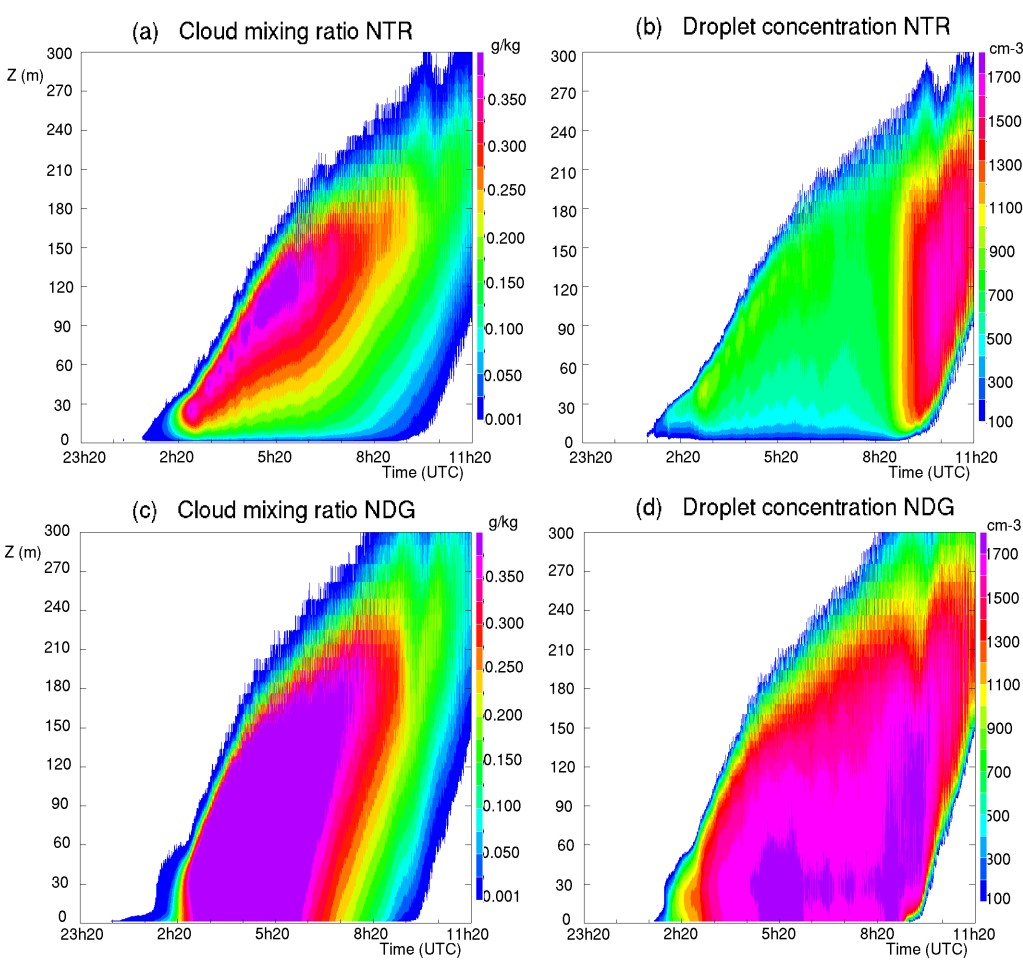

**Figure 11.** Temporal evolution of simulated vertical profiles of cloud mixing ratio (a and c, in $\mathrm{g\,kg^{-1}}$) and droplet concentration (b and d, in $\mathrm{cm^{-3}}$) for NTR and NDG simulations.





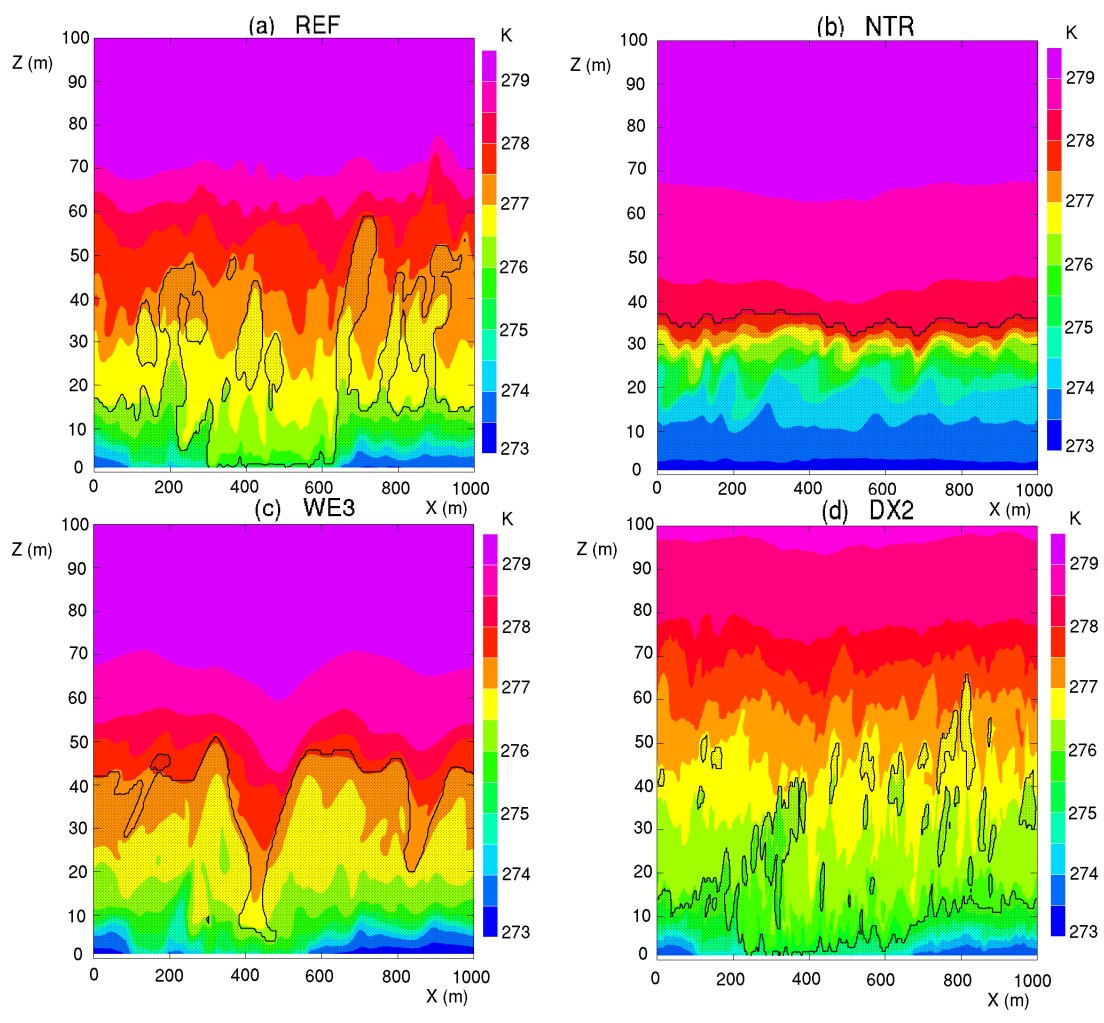

**Figure 12.** Vertical cross-sections at Y=500m and 0220 UTC of potential temperature (in $K$) for the REF (a), NTR (b), WE3 (c) and DX2 (d) simulations, with area of cloud mixing ratio higher than $0.1\,\mathrm{g\,kg^{-1}}$ superimposed with dots.





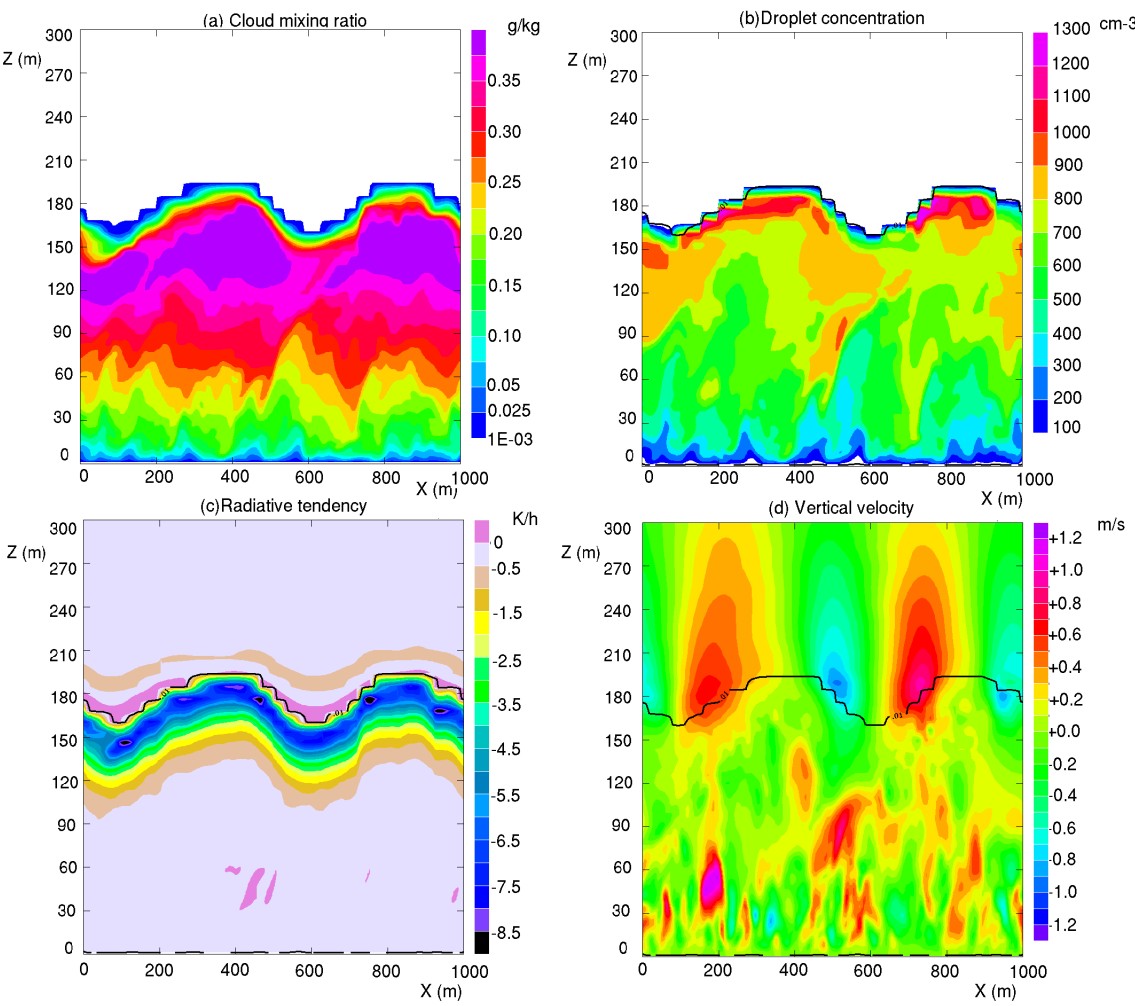

**Figure 13.** Vertical cross-section at Y=500m at 0620 UTC for the NTR simulation: (a) Cloud mixing ratio (in $g\,kg^{-1}$), (b) Droplet concen-centration (in $cm^{-3}$), (c) radiative tendency ( in K/h), (d) vertical velocity (in $m\,s^{-1}$) with the isoline of $r_c = 0.01\,g\,kg^{-1}$ superimposed.





**Figure 14.** Time series of droplet concentration (a, d and g, in $cm^{-3}$), liquid water content (b, e and h, in $g\,m^3$), and droplet size distribution (c, f and i, in $cm^{-3}$) at 0520 UTC and 3 m agl observed in black, and simulated in color.



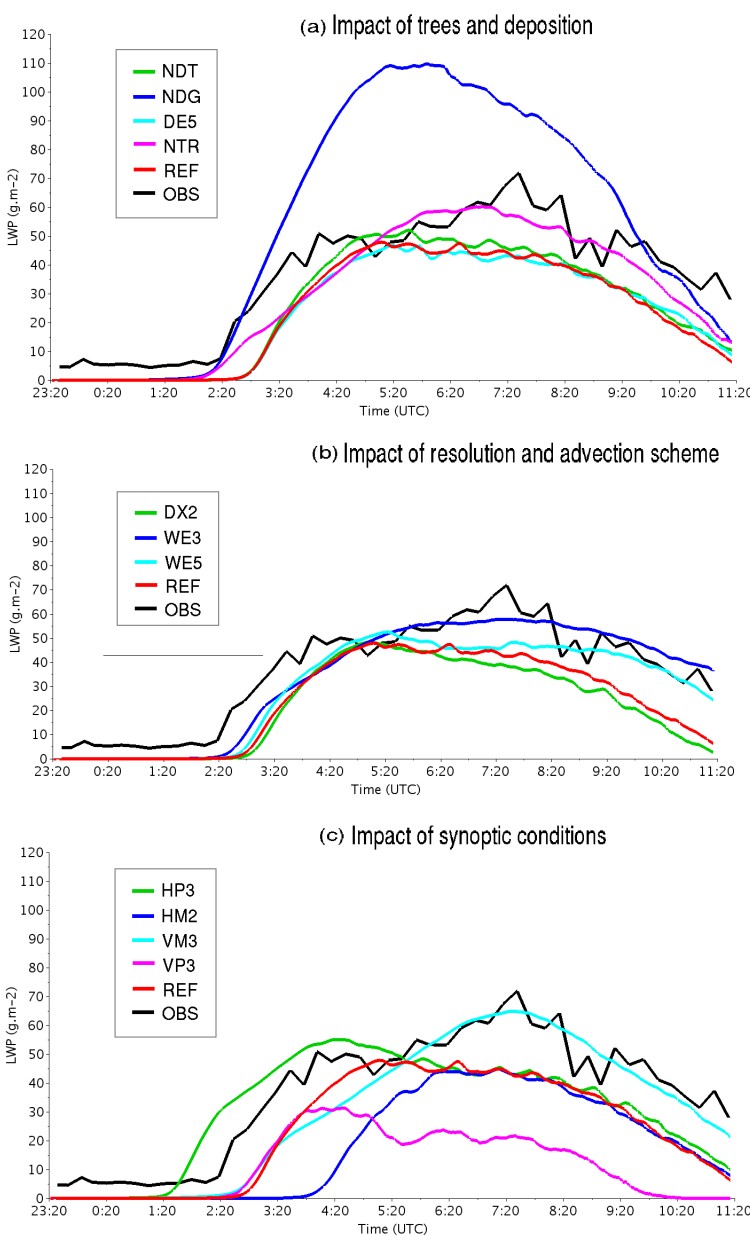

**Figure 15.** Time series of LWP (in $\mathrm{g\,m^{-2}}$) observed in black, and simulated in color for the different simulations.





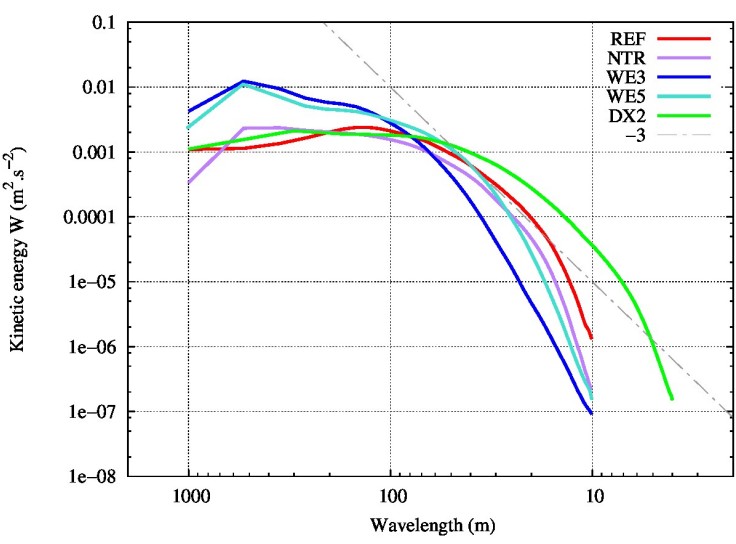

**Figure 16.** Mean kinetic energy spectra for vertical wind computed over the whole fog layer at 0620 UTC for the REF, WE3, WE5, DX2 and NTR simulations.



## [A] Appendix: Material support

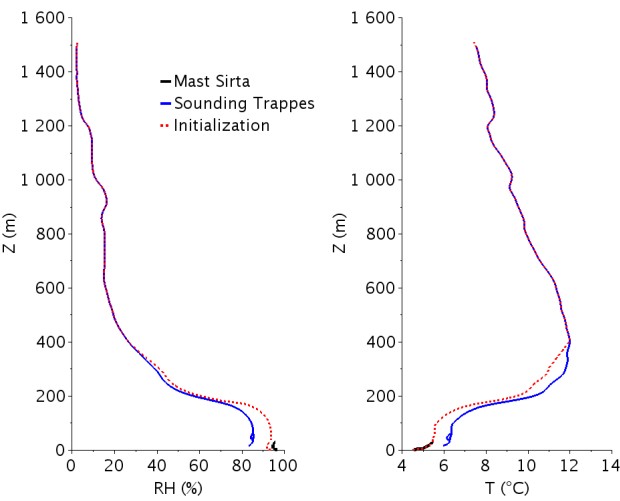

**Figure A.1.** Relative humidity (in %) and temperature (in $C$) vertical profiles at 2320 UTC on 14 November 2011 observed at the Sirta mast (in black), by the Trappes radiosounding (in blue) and used for the REF initialization.

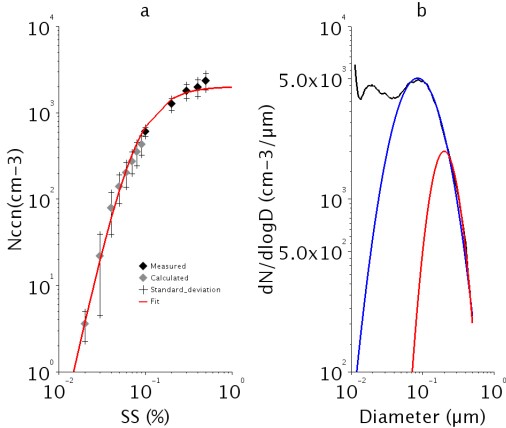

**Figure A.2.** (a) Activation spectrum: from CCNC measurement before the fog onset (between 0130 and 0230 UTC) for supersaturations higher than 0.1% in black dots, from calculation for supersaturations lower than 0.1% in grey dots, and fitted from the Cohard et al. (2000c) parametrization in red. (b) Particle size distribution from the aerosol measurements (in black), the lognormal distribution fitted on the accumulation mode (in blue) and according to Cohard et al. (2000c) (in red).





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
