# Peer review of "Large-eddy simulation of radiation fog: Part 1: Impact of dynamics on microphysics"

_Atmospheric Chemistry and Physics, 2016_

## Referee Comment (RC1) · Anonymous Referee #1 · 20 Nov 2016

Review of paper:

Large-eddy simulation of radiation fog: Part 1: Impact of dynamics on microphysics

By Mazoyer, Lac, Thouron, Bergot, Masson & Musson-Genon

Submitted to Atmospheric Chemistry & Physics

Manuscript: acp-2016-900

Recommended disposition: The manuscript requires major revisions before publication in Atmospheric Chemistry and Physics.

General comments:

The manuscript presents results from various LES of a radiation fog event observed at

a complex site. The simulations were aimed at identifying the main dynamical factors affecting the simulated life cycle of the fog layer, including its microstructure. The research is an original contribution toward a more complete understanding of the complex interactions shaping the evolution of a fog layer, as well as the identification of possible improvements of numerical models necessary for more accurate fog forecasts. The work is also a good example of how carefully crafted simulations can provide some insights into specific features often present in observations taken at complex sites. The discussion is comprehensive and generally well-structured, with major findings clearly emphasized. Some parts of the discussion could be shortened to further improve the clarity of the overall presentation. It is the opinion if this reviewer that major revisions are however needed before the manuscript can be accepted for publication.

More specifically:

1. First and foremost the written English is not of sufficient quality, which provides for a difficult read of the manuscript. It appears that the text was not put through a basic grammar check that most text editors have available. I highly encourage the authors to have the text revised by an English speaker to ensure appropriate terminology and sentence construction. There are also numerous opportunities to make the text more concise and clearer.

2. As this is a central aspect of this study, the parameterization of fogwater deposition on the tree canopy should be more clearly described and justified. In particular, the use of a drag term on momentum and TKE to represent the impact of a tree barrier on the flow and associated turbulence, while the use of a parameterization of fog water deposition which is entirely independent of the flow and turbulent characteristics (i.e. constant deposition velocity) may appear as incompatible. Hence, the chosen formulation should be more clearly justified and contrasted against the work of von Glasow and Bott (1999).

Specific comments:

1. Throughout the text, replace "trees barrier" by "tree barrier" or by "barrier of trees".

2. Use of past tense to describe some aspects of the simulations throughout the paper is awkward. You may have performed the simulations in the past, but their characteristics remain true now. Please revise your use of the past tense throughout the manuscript.

3. Throughout the manuscript, replace "ponctual" by "point".

4. Abtsract line 2: Revise with "during the ParisFog"

5. Abstract line 3: Please specify which aspect of "dynamics" you are referring to. Boundary layer?

6. Abstract line 5: deposition of what? Please specify for greater clarity.

7. Abstract line 7: We should read "as in observations" rather than "like in the observation".

8. Abstract last sentence: I would suggest re-wording as: …"necessary to accurately represent the fog life cycle at very high resolution" for a clearer statement.

9. Introduction line 18: How do you define "local dynamics"? and why do you not seem to include turbulence in that category?

10. Introduction line 19: Please rewrite with "understanding of fog processes" rather than "fog processes understanding".

11. Introduction, line 20: Sentence is without a verb.

12. Introduction, line 22: measurements (please use plural).

13. Introduction, line 22: "and set liquid water content": ? I do not understand. Please revise.

14. Page 2, line 5: use "as shown by Nakanishi"

15. Page 2, line 6: Here, need to add "to study some aspects of the characteristics of a fog layer". Nakanishi was not the first to use LES in general, as you seem to imply.

16. Page 2, line 8: "a turbulence scheme"

17. Page 2, line 11: Use of "stripes" is not appropriate. Maybe use "banded structures" and specify in which field(s) theses structures are observed.

18. Page 2, line 14: Replace "move" by "relocate".

19. Page 2, line 18: the word "Hence" is superfluous.

20. Page 2, line 26: The use of "allowing to represent" is not proper. Change to "allowing the representation of"

21. Page 2, line 30: replace "it" by "values"

22. Page 3, lines 3-4, sentence beginning with "Sensitivity tests will...": This has been said already. Please remove sentence.

23. Page 3, line 5: Replace "sophisticated microphysics" by "sophisticated microphysical parameterizarion scheme" to be more precise.

24. Page 3, line 6: Replace "taking into account" by "while accounting for".

25. Page 3, line 6: We should read "such as forests"

26. Page 3, line 14: winter of

27. Page 3, line 20: wind does not flow from a "side", rather from a direction. Also, "this side" implies that information about wind direction has been provided to the reader, which wasn't. Please revise your sentence(s).

28. Page 3, line 21: It is mentioned that the reader should refer to the study by Stolaki for a description of the instrumentation, yet the the entire next paragraph is devoted to just that. Please revise your text.

29. Page 3, line 23: Get rid of "At the surface". 30m is not "at the surface" in this context.

30. Page 3, line 31: We should read "Aerosol particle measurements", not "particles measurements"

31. Page 3, line 33: What type of profiler? I suppose it is a microwave profiler. Please be more precise with your statement.

32. Page 4, line 5: 1000 UTC "on the following morning"? Please be more precise.

33. Page 4, line 9: I am not clear as to why fog events were not classified as stratus lowering. 150m for initial cloud formation does seem high to be a radiation fog. Please explain.

34. Page 4, line 12: Replace "according to" by "following"

35. Page 4, line 14: Use of "moistening" could lead to confusion. Is "moistening" referring to an increase in *relative* humidity (due to cooling) or increase in absolute humidity (water vapor content)? Please be more precise.

36. Page 4, line 19: Not sure I understand the meaning of "temperature convergence" in this context.

37. Page 5, line 2: "fog droplet microphysics" is awkward wording in this context. Perhaps "fog microstructure" is more appropriate?

38. Page 5, line 5: "leaded" is not proper English.

39. Page 5, line 6: LWC and Nc decreased at 3m but visibility remained constant? Please explain.

40. Page 5, line 15: Can you be more precise in your description. Nut sure that "between" means in the context of a size distribution.

41. Page 5, line 29: "at the instrumental site"

42. Page 6, line 1: By "It" you mean "The drag approach"? Please be more precise.

43. Page 6, line 5: "a combination of the product" is confusing. A product already is a "combination" of terms. Simply say that it is a product of the fraction of vegetation with LAI and a weighting function. I would suggest that you show an equation for greater clarity and since it is a central aspect of your study.

44. Page 6, line 7: The vertical profile of what exactly? Please be more precise in you statement.

45. Page 6, line 7: "atlantic broad leaved trees": Where does that information included and how? Again, please be more precise in your statement. Perhaps refer to the equation that will show how Af is expressed.

46. Page 6, line 11: Aren't "activated CCN" droplets? Please clarify the difference between Nc and Nccn.

47. Page 7, line 16: How is droplet concentration and cloud mixing ratio taken into account in LW and SW calculations? Just provide appropriate references.

48. Page 8, line 2: I think here you rather mean that the **reduction** in visibility is underestimated. Please revise your statements.

49. Page 8, lines 8 and 9: .Variables are not transported. Perhaps simply write "momentum is advected"

50. Page 8, line 11: Awkward use of past tense.

51. Page 8, sentence on line 17-18: 1) soil moisture not moistening 2) Used the same point measurements to initialize soil variables across the entire domain? Please justify this approach.

52. Page 8, line 30, sentence with "good degree of confidence": This is not clearly justified. Please more directly and clearly address the possible shortcomings or impact of using this on your results. You should convince the reader that this mismatch does

not adversely affect your results.

53. Page 9, lines 3-4, "good degree of confidence" : Compared to surface observations? Please be more precise with your statement.

54. Page 9, statement on lines 5-6: making some assumption of ergodicity here? Taking time averages of point observations to compare to area-averaged simulated fields? Please describe more clearly the assumptions you are making and justify.

55. Page 10, line 22: What does "reducing the spectrum" mean? I do not know what a reduction in the spectrum mean.

56. Page 10, line 24: "leaded" is not proper English

57. Page 10, line 24: Awkward use of "weakness". Maybe replace by "underestimated"

58. Page 10, line 24: What do you mean by "surface cloud water amount by sedimentation"? Do you mean to say "amount of water deposited on the surface by sedimentation"?

59. Page 10, last sentence: Maybe an important point here about usefulness of more sophisticated formulations of visibility diagnostics for models. Your simulation results indicated that a simpler formulation based solely on LMC is adequate given the difficulty in simulating Nc. Perhaps this finding could be expanded upon here.

60. Page 11, line 15: "allows to decompose formally" is awkward. Maybe change to "serves as a basis for decomposing"

61. Page 11, line 19: "consecutively to the flow", you rather mean "related to the flow perturbations"?

62. Page 12, line 10, use of "rc" I believe you used "LWC" before. You should remain consistent throughout the paper.

63. Page 13, line 8: drawning? Please revise

Interactive
comment

64. Page 13, sentence on lines 10-11 is unclear. Please revise.

65. Page 13, line 31: statement with "even if measurements" is unclear. You mean "...probably overestimated, although this cannot be confirmed as measurements ..."

66. Page 14, sentence on lines 24-25 is confusing. Please revise.

67. Page 16, line 5: "removed fully deposition" should be replaced by "removed deposition altogether" for proper wording.

68. Page 16, line 18, LWP was largely overestimated. Where? At the surface? If so, how is LWC at surface positively correlated to the depth of the fog layer? Please provide a clearer explanation.

69. Page 16, line 21: Is DE5 based on deposition on a grassy surface only, or is deposition over the entire tree canopy considered as well? In the context of this section, this text is not clear. Please clarify.

70. Page 16, line 21: Why not use a value of 8 cm s-1, the upper bound suggested by Katata?

71. Page 16, line 22: Replace "diminution" by "reduction".

72. Page 16, line 29: Replace "the remove of" by "neglecting" for proper wording.

73. Page 17, line 26: I do not think "preformation" is a word. Maybe you mean "initial formation"?

74. Page 17, line 27: A DSD does not "move". Maybe "characterized by higher concentrations of larger droplets"

75. Page 17, line 30: "dilutes" is not properly used here. You rather mean "decreases" or "diminishes".

76. Page 17, line 30: Also this reduced effect impacts which field(s) in particular. Please clarify.

[Figure]

77. Page 17, line 32, "fog slightly deeper": Please revise as "a slightly deeper fog layer"

78. Page 18, lines 26-27: I do not understand the statement "diverged on the fog life cycle in the same way". Please revise your statement.

79. Page 18, lines 27-28: Not a very clear statement. Please revise. And be more explicit about what you mean by "dynamical conditions".

80. Page 19, line 10, "as the wind overcame this obstacle": Awkward formulation. Maybe "and associated perturbed mean flow and turbulence conditions" would be a clearer statement.

81. Page 19, line 17: replace "meeting" with "encountering" or "reaching".

82. Page 19, line 17: use of the expression "dynamical gradients" is not specific enough. Do you mean "wind shear" in particular?

83. Page 19, line 18, "became well-marked": This is awkward wording. Do you mean "became prominent"?

84. Page 19, line 23,"homogeneous". Where? Throughout the fog layer? At the top of the layer? Please be more precise in your statement.

85. Page 19, line 24: "evolved" rather than "involved"?

86. Page 19, line 29: "damaging the visibility diagnostic" is awkward wording. Maybe "worsening visibility diagnostics"?

87. Page 19, line 31: "The removal of the deposition process" is awkward wording. Maybe replace "The removal of" by "Neglecting".

88. Page 20, line 4, "Endly": You mean "Finally" or "Lastly"?

89. Page 20, lines 4-5: The use of "reduce much more the number concentration" is awkward. Change to "reduce the overestimated droplet number concentration" for a more precise statement.

90. Page 20, line 8: In what way "simulations remain very challenging"? Please explain.

91. Page 20, line 12: I suggest you replace "cannot be neglected anymore" by "should not be neglected"

92. Page 20, line 20: We should read "dewfall" instead of "dewfal"

93. Page 20, line 23: Change "no one" to "none" for appropriate wording.

94. Page 20, line 23: Change "to reproduce correctly" to "in correctly reproducing"

95. Page 39: The citation of Hammer is not accurate. That paper has now been fully published and the citation should now indicate : Atmos. Chem. Phys., 14, 10517-10533, doi:10.5194/acp-14-10517-2014

96. The formatting of citations is inconsistent throughout the References section. In particular, the names of journals sometimes uses capital letters (as should be) and sometimes not. Please revise.

Note: Only the most important text corrections have been suggested. A much greater number of possible corrections have been omitted due to time constraints for the reviewer. I strongly recommend that the text be reviewed by someone with a higher proficiency in English.
* * *

---

## Referee Comment (RC2) · Anonymous Referee #2 · 24 Nov 2016

Review of "Large-eddy simulation of radiation fog: Part 1: Impact of dynamics on microphysics" by By Mazoyer, Lac, Thouron, Bergot, Masson & Musson-Genon Submitted to Atmospheric Chemistry & Physics Manuscript: acp-2016-900

Recommendation: Major revisions required.

Overview: This manuscript presents an original and thorough examination of a fog event at a site with varying land use (grass and trees). The authors have used LES simulations to assess the impact of a line of trees on the formation and lifecycle of the fog. A variety of different simulations were used to determine which processes were having the largest effects on the fog, and this has resulted in improved understanding of this scenario, as well as some recommendations for improvements in further fog simulations.

[Figure]

The presentation of the manuscript could be significantly improved by being proofread by a fluent or native English speaker. There are many spelling and grammatical errors in the manuscript, as well as a considerable number of instances of awkward phrasing. The formatting of the manuscript is also inconsistent. These problems make the manuscript very hard work to read, and obscures the nuance and scientific value of the authors work, which is otherwise good.

Scientific comments:

1. P3, line 33: What sort of profiler are you using?

2. P4, line 17: How do you differentiate between radiation fog forming under very low (150m) cloud, and cloud lowering to the surface? You describe this event as follows: "the cloud base height progressively subsided during about 30 min, until it reached the ground". This sounds indistinguishable from stratus fog.

3. P4, lines 24 & 25: It is not clear to which TKE measurement you are referring here. The increase in TKE at 10 m occurs ∼30 minutes before the increase at 30 m, not simultaneously. After this increase there is still quite a lot of variability in the TKE, so I would not describe it as constant.

4. P5, lines 1 & 2: There is a ∼30 minute difference in timing between the increase in LWC and Nc.

5. P9, line 13: More detail about the temperature convergence is required – i.e. the temperatures measured at different heights converge.

6. P9, line 14: If only RH is being considered, it is not accurate to say that fog formed at 0230, only that saturation was reached. You need to refer to e.g. a visibility measurement.

7. P9, line 22: This increase in TKE occurs > 30 minutes before the TKE increase in the observations.

[Figure]

8. P10, line 25: Please define the difference between sedimentation and deposition.

9. P11, line 9 and onwards: You keep switching between LWC and rcǍň throughout the manuscript. It would be better to consistently use one or the other.

10. There are a few statements throughout the manuscript which are accompanied by "not shown". Is there a particular reason why they are mentioned, but not included in plots?

11. P15, line 6: What do you mean by the "production" of Nc?

12. P16, line 5: In the context of fog microphysics, 3m is not especially "near surface".

Technical comments:

1. Section 2.3.1: Not all terms of the equations presented in this section are defined in the text.

2. Section 2.2.3: The figure numbers in this section do not correspond to any of the figure captions.

3. P9, lines 17-19: It is difficult to see the negative temperature gradients in Fig. 2, due to the number of lines.

4. P9, lines 20 & 22: Please refer to Fig. 3a & 3b, instead of just Fig. 3.

5. P11, lines 15-17: It would be helpful to the reader if the different phases of the fog lifecycle were marked on any plots showing a time series of data.

6. P11, line 30: "when the fog reached approximately 80 m". Is this the depth of the fog, the height of the fog top, or the location of the cloud/fog base?

7. P11, line 33: Fig. 7d shows updraft velocity, not cooling.

8. Section 3.3: Marking the location of the trees on plots of spatially varying data would make the figures easier to interpret.

[Figure]

9. Please put the figures in the order in which they are first referred to in the text.

10. There are numerous occasions where the figures are incorrectly referenced in the text. Please correct this.

11. When plotting a time series from the LES, please state where in the domain the data was from.

12. P18, line 12: Are you referring here to the surface, or 3m?

13. References: The capitalisation of journal titles and place names in the reference list is inconsistent, there are also some references missing page numbers.

---

## Referee Comment (RC3) · Anonymous Referee #3 · 29 Nov 2016

This paper presents large eddy simulations of a radiation fog event for which extensive research quality observations were available. The main focus of the paper is to uncover how different aspects of the model dynamics affect the fog evolution, and sensitivity to the surface treatment, initial conditions and model dynamical formulation are investigated. Whilst the work is interesting, and ultimately worthy of publication, I feel extensive modifications to the manuscript are required before it is suitable for publication.

Firstly, the manuscript is very difficult to read, due to numerous spelling and grammatical mistakes. A revised version would benefit from extensive proof-reading and typographical editing, possibly with the help of a native English speaker. I have provided suggestions for the abstract below, to give the authors an idea on the level of modification required:

L2 - should say "...during the ParisFog..."

L4 - should say "...of a tree barrier..."

L7 - should say "...as in the observations, and..."

L10 - should say "...meaning that grid convergence..."

L12 - should say "...and had a similar effect to removing the tree barrier."

L13 - should say "...allows us to..."

L13 - should say "...necessary to correctly simulate the fog life cycle at high resolution."

Secondly, the manuscript lacks structure and coherence. It currently just presents a long list of things you have done, with no real theme linking everything together or justifying the various experiments. The introduction should focus on the specific problem you are trying to address - how dynamics affects the evolution of fog, what specific questions are you trying to answer? This should then provide justification for the sensitivity experiments you conduct - how do they help you answer the questions? The conclusions should then tie all this together and answer those questions. It is possible that in doing this, you may be able to shorten the text (which is currently quite long) and number of figures, to only focus on what is really relevant.

I only have two specific scientific comments:

Sect 2.3.2 - why do you choose an empirical diagnosis of visibility based on the cloud water content and drop number, rather than calculating the visibility accurately from Eqn. 7? With the complicated microphysics scheme you have available, you should be able to calculate the extinction coefficient directly, e.g. as done by Clark et al. (2008).

P9, L32 - do you have observations of the surface or soil temperatures which you could compare to the model here to explain the difference in upwelling LW radiation?

Reference:

Clark, P. A., Harcourt, S. A., Macpherson, B., Mathison, C. T., Cusack, S. and Naylor, M. (2008), Prediction of visibility and aerosol within the operational Met Office Unified Model. I: Model formulation and variational assimilation. Q.J.R. Meteorol. Soc., 134: 1801–1816. doi:10.1002/qj.318
* * *

---

## Author Comment (AC1) · 31 Jan 2017

Dear Sir,

We would like to thank you sincerely for your precious support to correct the text, and all your suggestions. Before answering to your questions, we must confess that there was an error in the coding of the deposition processă: the deposition velocity was mistakenly multiplied by the volume of the grid, corresponding to a ratio of 25 for all the simulations at 5m resolution (so a deposition velocity of 50 cm/s instead of 2 cm/s was actually applied), and to a ratio of 4 for the simulation at 2m resolution (noted DX2). Consequently, the deposition effect was overestimated. All the simulations except the one without deposition (called NDG) have been run again and most of the figures have been updated. For the REF simulation (with a deposition velocity of 2 cm/s), the dis-

crepancies with the observed microphysical fields are a bit stronger (cloud mixing ratio and droplet concentration more overestimated), but the DE8 simulation (deposition velocity of 8 cm/s as you proposed) presents a significant improvement. The signature of the fog onset at elevated levels in the REF simulation is not so marked, and is more evident in the DE8 simulation, showing that both the tree drag effect and the deposition are necessary to reproduce the formation of fog at elevated levels. The new results do not modify the analysis of the fog event and the conclusions of the study. The text has been also reduced to answer to the reviewersÂă: the sensitivity test on the initial conditions has been removed, as well as the corresponding figures. The length of the text has been reduced as expected. Lastly, the text has been revised by an english native speaker.

Best regards,

Christine Lac

Please also note the supplement to this comment:
http://www.atmos-chem-phys-discuss.net/acp-2016-900/acp-2016-900-AC1-supplement.pdf

**Supplement:**

We would like to thank you sincerely for your precious support to correct the text, and all your suggestions. Before answering to your questions, we must confess that there was an error in the coding of the deposition process : the deposition velocity was mistakenly multiplied by the volume of the grid, corresponding to a ratio of 25 for all the simulations at 5m resolution (so a deposition velocity of 50 cm/s instead of 2 cm/s was actually applied), and to a ratio of 4 for the simulation at 2m resolution (noted DX2). Consequently, the deposition effect was overestimated.
All the simulations except the one without deposition (called NDG) have been run again and most of the figures have been updated. For the REF simulation (with a deposition velocity of 2 cm/s), the discrepancies with the observed microphysical fields are a bit stronger (cloud mixing ratio and droplet concentration more overestimated), but the DE8 simulation (deposition velocity of 8 cm/s as it was requested by one of the reviewers) presents a significant improvement. The signature of the fog onset at elevated levels in the REF simulation is not so marked, and is more evident in the DE8 simulation, showing that both the tree drag effect and the deposition are necessary to reproduce the formation of fog at elevated levels. The new results do not modify the analysis of the fog event and the conclusions of the study.
The text has been also reduced to answer to the reviewers : the sensitivity test on the initial conditions has been removed, as well as the corresponding figures. The length of the text has been reduced as expected. Lastly, the text has been revised by an english native speaker.

*Recommended disposition: The manuscript requires major revisions before publication in Atmospheric Chemistry and Physics.*

*General comments:*

*The manuscript presents results from various LES of a radiation fog event observed at a complex site. The simulations were aimed at identifying the main dynamical factors affecting the simulated life cycle of the fog layer, including its microstructure. The research is an original contribution toward a more complete understanding of the complex interactions shaping the evolution of a fog layer, as well as the identification of possible improvements of numerical models necessary for more accurate fog forecasts. The work is also a good example of how carefully crafted simulations can provide some insights into specific features often present in observations taken at complex sites. The discussion is comprehensive and generally well-structured, with major findings clearly emphasized. Some parts of the discussion could be shortened to further improve the clarity of the overall presentation. It is the opinion if this reviewer that major revisions are however needed before the manuscript can be accepted for publication.*

*More specifically:*
*1. First and foremost the written English is not of sufficient quality, which provides for a difficult read of the manuscript. It appears that the text was not put through a basic grammar check that most text editors have available. I highly encourage the authors to have the text revised by an English speaker to ensure appropriate terminology and sentence construction. There are also numerous opportunities to make the text more concise and clearer.*

Additionally to all your suggestions, the text has been revised by a native speaker of English.

*2. As this is a central aspect of this study, the parameterization of fogwater deposition on the tree canopy should be more clearly described and justified. In particular, the use of a drag term on momentum and TKE to represent the impact of a tree barrier on the flow and associated turbulence,*

*while the use of a parameterization of fog water deposition which is entirely independent of the flow and turbulent characteristics (i.e. constant deposition velocity) may appear as incompatible. Hence, the chosen formulation should be more clearly justified and contrasted against the work of von Glasow and Bott (1999).*

You are right that the tree drag parametrization is sophisticated while the parameterization of fogwater deposition is simplistic.The tree drag parametrization has been introduced quite a long time ago in the model (Aumond et al., 2013) and validated on different cases (Bergot et al., 2015a). On the contrary, the deposition process is not taken into account in most of the models, especially NWP models. The first step here is therefore to have a first approach by examining the importance of this process, considering a simplistic formulation. As the conclusion is that this term is essential to correctly reproduce microphysical fields of the fog cycle, the next step in a further study will be to have a more sophisticated formulation as in von Glasow and Bott (1999). The text has been modified like this :

« In addition to droplet sedimentation, fog deposition is also introduced which represents direct droplet interception by the plant canopies. In the real world, it results from turbulent exchange of fog water between the air and the surface underneath, leading to collection (Lovett et al., 1997). In numerical weather prediction models (NWP), this process is most of the time not included, such as in the French NWP model AROME (Seity et al., 2011) whose physics comes from Meso-NH. As a new process to introduce, only a simple formulation of the deposition process is considered here as a first step, in order to perform a sensitivity study. The fog deposition flux $F_{DEP}$ is predicted at the first level of the atmospheric model (50 cm height) for grassy areas, and over the 15 m height for trees, in a simplistic way following Zhang et al. (2014b): $F_{DEP} = aV_{DEP}$ with $= r_c;N_c$ and where $V_{DEP}$ is the deposition velocity. In a review based on measurements and parametrizations, Katata (2014) showed that $V_{DEP}$ values ranged from 2.1 to 8.0 cm/s for short vegetation. Here $V_{DEP}$ is assumed to be constant, equal to 2 cm/s. A test of sensitivity to this value is presented below. Water sedimentation and deposition amounts are input to the humidity storage of the surface model. A more complete approach in a further study would include a dependance of $V_{DEP}$ on momentum transport as in von Glasow and Bott (1999) and also on LAI.»

The simplistic formulation of the deposition process and the necessity to improve it was already underlined in the conclusion : « In this study, the deposition term was introduced quite crudely and this would need some refinements in further studies. It would need to take account of the wind speed and the turbulence , and it could also consider the hygroscopic nature of canopies. By analogy with dry deposition, it would also be better to take droplet diameter into account, assuming that this field is correctly reproduced. Other studies have also shown that fog water deposition is strongly enhanced at the forest edge, becoming up to 1.5-4 times larger than that in closed forest canopies (Katata, 2014), so it could be interesting to simulate the edge effect of fog water deposition. ».

*Specific comments:*
 1. *Throughout the text, replace "trees barrier" by "tree barrier" or by "barrier of trees".*
    OK

 2. *Use of past tense to describe some aspects of the simulations throughout the paper is awkward. You may have performed the simulations in the past, but their characteristics remain true now. Please revise your use of the past tense throughout the manuscript.*
    OK

 3. *Throughout the manuscript, replace "ponctual" by "point".*
    OK

 4. *Abtsract line 2: Revise with "during the ParisFog"*
    OK

5. *Abstract line 3: Please specify which aspect of "dynamics" you are referring to Boundary layer?*
   Yes, it has been corrected by « the dynamics of boundary layer »

6. *Abstract line 5: deposition of what? Please specify for greater clarity.*
   Yes, « deposition of droplets »

7. *Abstract line 7: We should read "as in observations" rather than "like in the observation".*
   OK

8. *Abstract last sentence: I would suggest re-wording as: : : :"necessary to accurately represent the fog life cycle at very high resolution" for a clearer statement.*
   Yes, thank you.

9. *Introduction line 18: How do you define "local dynamics"? and why do you not seem to include turbulence in that category?*
   I mean by local dynamics local flow due to orography for instance. This has been corrected by « local flow »

10. *Introduction line 19: Please rewrite with "understanding of fog processes" rather than "fog processes understanding".*
    Yes.

11. *Introduction, line 20: Sentence is without a verb.*
    OK, « can be referred » has been added.

12. *Introduction, line 22: measurements (please use plural).*
    OK

13. *Introduction, line 22: "and set liquid water content": ? I do not understand. Please revise.*
    OK, set has been replaced by report.

14. *Page 2, line 5: use "as shown by Nakanishi"*
    OK

15. *. Page 2, line 6: Here, need to add "to study some aspects of the characteristics of a fog layer". Nakanishi was not the first to use LES in general, as you seem to imply.*
    OK, this has been corrected by : « Many important features of fog have also been characterized using one-dimensional (1D) modelling (Bergot et al. (2007), Tardif (2007), Stolaki et al. (2015) among others). However, to study some aspects of the characteristics of a fog layer, it has become necessary to explicitly simulate turbulence motions in 3D as shown by Nakanishi (2000) who was the first to use a large-eddy simulation (LES) for fog. »

16. *Page 2, line 8: "a turbulence scheme"*
    OK

17. *Page 2, line 11: Use of "stripes" is not appropriate. Maybe use "banded structures" and specify in which field(s) theses structures are observed.*
    OK, but « stripes » was already used by Bergot (2013). This has been corrected by : « During the formation phase, small banded structures, identified by Bergot(2013) as Kelvin-Helmotz (KH) billows, occur in the middle of the fog layer on dynamical and thermodynamical fields. »

*18. Page 2, line 14: Replace "move" by "relocate".*
OK

*19. Page 2, line 18: the word "Hence" is superfluous.*
OK

*20. Page 2, line 26: The use of "allowing to represent" is not proper. Change to "allowing the representation of"*
OK

*21. Page 2, line 30: replace "it" by "values"*
OK

*22. Page 3, lines 3-4, sentence beginning with "Sensitivity tests will: : :": This has been said already. Please remove sentence.*
The sentence has been removed.

*23. Page 3, line 5: Replace "sophisticated microphysics" by "sophisticated microphysical parameterizarion scheme" to be more precise.*
OK

*24. Page 3, line 6: Replace "taking into account" by "while accounting for".*
OK

*25. Page 3, line 6: We should read "such as forests"*
OK

*26. Page 3, line 14: winter of*
OK

*27. Page 3, line 20: wind does not flow from a "side", rather from a direction. Also, "this side" implies that information about wind direction has been provided to the reader, which wasn't. Please revise your sentence(s).*
OK, this has been corrected by : : « Zaïdi et al. (2013) demonstrated the impact of the tree barrier on the observed flow when the wind was blowing from this direction, and our case study was in this configuration. »

*28. Page 3, line 21: It is mentioned that the reader should refer to the study by Stolaki for a description of the instrumentation, yet the the entire next paragraph is devoted to just that. Please revise your text*
The reference related to Stolaki's study for the description of instrumentation has been removed.

*29. Page 3, line 23: Get rid of "At the surface". 30m is not "at the surface" in this context.*
« At the surface » has been removed.

*30. Page 3, line 31: We should read "Aerosol particle measurements", not "particles measurements"*
OK

*31. Page 3, line 33: What type of profiler? I suppose it is a microwave profiler. Please be more precise with your statement.*

Yes, this is a RPG-HATPRO water vapour and oxygen multi-channel microwave profiler : this information has been added.

32. *Page 4, line 5: 1000 UTC "on the following morning"? Please be more precise.*
It has been added.

33. *Page 4, line 9: I am not clear as to why fog events were not classified as stratus lowering. 150m for initial cloud formation does seem high to be a radiation fog. Please explain.*
You are right that the distinction between radiative fog and cloud lowering is not easy to make. Fog classifications traditionally use the Tardif and Rasmussen (2007) method. They differentiate stratus lowering from radiative fog by the wind speed and the cloud ceiling. If the wind speed at 10m is lower than 2.5 cm/s before the formation and the cloud ceiling is less than 100m then the fog is supposed to be radiative. Our measured wind speed at 10m is under 2.5cm/s but our cloud ceiling is higher than 100m (150m). However according to Dupont et al. (2012), the lowering of a stratus is due to a cooling at its base by evaporation of sedimented droplets. Considering fall speed of 2.2 cm/s (Roach et al, 1976) it would necessitate at less 10 hours for the cloud to reach the ground. Moreover we do believe that the cloud formation at 150m is due to the modification of the flow caused by the tree barrier resulting in an important vertical mixing on a significant depth. So we conclude that this fog is a radiative one.
We propose the text :
« As underlined by Stolaki et al. (2015), this characteristic is very common at Sirta and 88% of the radiation fog events during the field experiment were also elevated. However, they were not classified as stratus lowering as they were followed rapidly by formation of fog at the surface. A delay of 30 min between the formation at 150 m height and at the ground seems too short to be a stratus lowering, which is mainly driven by the evaporation of slowly falling droplets that cool the sub-cloud layer (Dupont et al., 2012). This suggests that this type of radiation fog could be linked with, and specific to, the configuration of the Sirta site. »

34. *Page 4, line 12: Replace "according to" by "following"*
OK

35. *Page 4, line 14: Use of "moistening" could lead to confusion. Is "moistening" referring to an increase in \*relative\* humidity (due to cooling) or increase in absolute humidity (water vapor content)? Please be more precise.*
You are right that it was confusing. The increase in relative humidity is associated to the cooling, as we can see below on the dewpoint temperature : the difference between temperature and dewpoint temperature reduces slowly until the fog formation. « as well as a moistening » has been replaced by «inducing an increase in relative humidity ».

[Figure]

*Figure : Temporal evolution of observed relative humidity (a) and temperature and dewpoint temperature (b) from 10 UTC the 14th of November to 12 UTC the 15th.*

36. *Page 4, line 19: Not sure I understand the meaning of "temperature convergence" in this context.*
    This has been corrected by : « At 0230 UTC, the apparition of fog at the ground was associated with a temperature homogenization in the first 30 metres, called temperature convergence by Price (2011) and corresponding to a neutral layer.»

37. *Page 5, line 2: "fog droplet microphysics" is awkward wording in this context. Perhaps "fog microstructure" is more appropriate?*
    « liquid droplet » has been removed.

38. *Page 5, line 5: "leaded" is not proper English.*
    OK, it has replaced by « brought ».

39. *Page 5, line 6: LWC and Nc decreased at 3m but visibility remained constant? Please explain.*
    In fact, LWC and Nc decrease but visibility increases slightly. This has been corrected.

40. *Page 5, line 15: Can you be more precise in your description. Not sure that "between" means in the context of a size distribution.*
    OK, this has been corrected by : « During the dissipation phase (in green, at 0700 UTC), the concentration of larger droplets fell but remained higher than initially. »

41. *Page 5, line 29: "at the instrumental site »*

OK

42. *Page 6, line 1: By "It" you mean "The drag approach"? Please be more precise.*
Yes : « The drag approach  consists of introducing an additional term in the momentum and TKE equations »

43. *Page 6, line 5: "a combination of the product" is confusing. A product already is a "combination" of terms. Simply say that it is a product of the fraction of vegetation with LAI and a weighting function. I would suggest that you show an equation for greater clarity and since it is a central aspect of your study.*
The sentence has been corrected but we have not introduced an additional equation as the formulation is exactly described by the sentence : «  $A_f$ ( z) is the product of the fraction of vegetation in the grid cell by the leaf area index (LAI) and by a weighting function representing the shape of the trees, as presented in Aumond et al. (2013). »

44. *Page 6, line 7: The vertical profile of what exactly? Please be more precise in you statement.*
Yes, this has beeen corrected : see 43.

45. *Page 6, line 7: "atlantic broad leaved trees": Where does that information included and how? Again, please be more precise in your statement. Perhaps refer to the equation that will show how Af is expressed.*
This information does not refer to the weighting function but to the vegetation cover introduced for the trees, which will be considered by the land surface scheme. This has been clarified by : « The trees introduced in the simulation domain for the land surface scheme correspond to Atlantic coast broad leaved trees » instead of « We have considered atlantic coast broad leaved trees ».

46. *Page 6, line 11: Aren't "activated CCN" droplets? Please clarify the difference between Nc and Nccn.*
At the beginning, concentrations of activated CCN and droplets are equal but then droplet concentration is modified by several mechanisms as break-up, evaporation, autoconversion, accretion and sedimentation so concentrations of activated CCN and droplets become different. This point is presented in the 2 references relative to the microphysical scheme : Khairoutdinov and Kogan (2000) and Geoffroy et al. (2008).
Another point is that according to the Köhler theory, for a given maximum supersaturation Smax, aerosols activated are exactly those with a critical supersaturation lower than Smax. Thus, to determine the number of aerosols really activated at time t, we first compute the number of activable aerosols for Smax. The number of aerosols really activated is then the difference between the number of activable aerosols and the number of aerosols previously activated during the simulation. This point has been added.

47.  *Page 7, line 16: How is droplet concentration and cloud mixing ratio taken into account in LW and SW calculations? Just provide appropriate references.*
The radiative transfer is computed with the ECMWF radiation code, using the Rapid Radiation Transfer Model (RRTM, Mlawer et al. (1997)) for longwave and Morcrette (1991) for shortwave radiation. Cloud optical properties for LW and SW radiation take account of the cloud droplet concentration in addition to the cloud mixing ratio. For SW radiation, the effective radius of cloud particle is calculated from the 2-moment microphysical scheme, the optical thickness is parametrized according to Savijärvi et al. (1997), the asymetry factor from Fouquart et al. (1991) and the single scaterring albedo from Slingo (1989). For LW radiation, cloud water optical properties refer to Savijärvi et al. (1997).

48. *Page 8, line 2: I think here you rather mean that the \*\*reduction\*\* in visibility is*

*underestimated. Please revise your statements.*
No, on Fig.6, the green curve is below the black one ; the parametrized visibility according to Zhang underestimates the observed visibility.

49. *Page 8, lines 8 and 9: .Variables are not transported. Perhaps simply write "momentum is advected"*
OK

50. *Page 8, line 11: Awkward use of past tense.*
The past tense throughout the manuscript has been replaced by the present tense.

51. *Page 8, sentence on line 17-18: 1) soil moisture not moistening*
OK
*2) Used the same point measurements to initialize soil variables across the entire domain? Please justify this approach.*
Soil measurements are available in one place. As we consider a flat terrain and only two cover types (grass and trees) in the simulation, it makes sense.

52. *Page 8, line 30, sentence with "good degree of confidence": This is not clearly justified. Please more directly and clearly address the possible shortcomings or impact of using this on your results. You should convince the reader that this mismatch does not adversely affect your results.*
A representation of the activation is a crucial point of this study as it directly links the calculated supersaturation to activated aerosol concentration. Usually, to find the Cohard et al. (2000c) parameter values, a fit is made on the aerosol lognormal distribution. Thanks to the CCNC, we get the exact curve of the evolution of the activated aerosol concentration, but only for supersaturation above 0.1 percent. As the activation in a fog layer is supposed to be under 0.1 percent, an instrumental method has been developed by Mazoyer et al. (2016) to retrieve the activation spectrum under this value. Using the combination of both information provides the exact activation spectrum, meaning that there is no shortcoming to use this method.
We have addressed this point more directly :
« Nevertheless, considering that the activation spectrum is deduced from measurements, it includes a good degree of confidence. » has been replaced by : « Deducing the activation spectrum  from measurements provides the exact solution. »

53. *Page 9, lines 3-4, "good degree of confidence" : Compared to surface observations? Please be more precise with your statement.*
«degree of confidence » has been replaced by « agreement with observation »

54. *Page 9, statement on lines 5-6: making some assumption of ergodicity here? Taking time averages of point observations to compare to area-averaged simulated fields? Please describe more clearly the assumptions you are making and justify.*
No, it does not correspond to some assumption of ergodicity. The horizontal variability study (Fig.9a for instance) shows that the domain near the surface can be decomposed into 4 meriodional bands with similar characteristics inside each one : the first one upstream from the trees, the second one corresponding to the barrier, the third one downstream the trees and the last one far downstream the trees. The instrumented area is located inside the third one so we have averaged the simulated fields on this band to compare to the measurements.
You are right that the sentence was not clear. We propose :
« It should be emphasized that observations localized at one point will be compared to simulated fields averaged over a horizontal area located downstream of the tree barrier (blue contour area of Fig. 1b) representative of the instrumented area.  We will indeed

see that the simulation domain is divided into 4 parts with significant differences between them, but similar characteristics inside each one.»

55. *Page 10, line 22: What does "reducing the spectrum" mean? I do not know what a reduction in the spectrum mean.*
We wanted to say that the number of larger droplets has been reduced. This part has been simplified and adapted to the new results : « During the whole fog life cycle, the model overestimates droplets with a diameter larger than 4 m  and underestimates the smaller ones.»

56. *Page 10, line 24: "leaded" is not proper English*
OK.

57. *Page 10, line 24: Awkward use of "weakness". Maybe replace by "underestimated"*
OK.

58. *Page 10, line 24: What do you mean by "surface cloud water amount by sedimentation"? Do you mean to say "amount of water deposited on the surface by sedimentation"?*
Yes. This part has been simplified and the new comment is :  « The cloud water deposition rate at the ground presents a maximum of 0.36 mm/day while the maximum of droplet sedimentation rate is 0.08 mm/day, meaning that the deposition is the main contributor to the cloud water amount at the ground. »

59. *Page 10, last sentence: Maybe an important point here about usefulness of more sophisticated formulations of visibility diagnostics for models. Your simulation results indicated that a simpler formulation based solely on LMC is adequate given the difficulty in simulating Nc. Perhaps this finding could be expanded upon here.*
Thank you. It has been added : « This explains why a simpler formulation of visibility based solely on $r_c$  is usually more adequate given the difficulty of simulating $N_c$  for the models.»

60. *Page 11, line 15: "allows to decompose formally" is awkward. Maybe change to "serves as a basis for decomposing"*
Thank you.

61. *Page 11, line 19: "consecutively to the flow", you rather mean "related to the flow perturbations"?*
No, we just mean that the layer of  TKE deepens slowly due to the tree barrier. It has been corrected.

62. *Page 12, line 10, use of "rc" I believe you used "LWC" before. You should remain consistent throughout the paper.*
Yes, we agree. Only cloud mixing ratio is now only used throughout the paper.

63. *Page 13, line 8: drawning? Please revise*
Yes, replaced by « bringing »

64. *Page 13, sentence on lines 10-11 is unclear. Please revise.*
«The fog forms at the surface upstream from the trees, and 500 m downstream, while it appears first at elevated levels between both » has been replaced by « The fog forms at the surface upstream of the trees, and 500 m far downstream, while it appears first at elevated levels over the intermediate area between the trees and far downstream (Fig. 9d).»

65. *Page 13, line 31: statement with "even if measurements" is unclear. You mean"...probably overestimated, although this cannot be confirmed as measurements ..."*
Yes, thank you.

66. *Page 14, sentence on lines 24-25 is confusing. Please revise.*
We propose : « The main differences in dynamics between NTR and REF appear  first on total TKE, with a thinner layer of TKE values higher than 0.5 m²/s² and smaller maxima  (Fig. 8b). »

67. *Page 16, line 5: "removed fully deposition" should be replaced by "removed deposition altogether" for proper wording.*
Yes, thank you.

68. *Page 16, line 18, LWP was largely overestimated. Where? At the surface? If so, how is LWC at surface positively correlated to the depth of the fog layer? Please provide a clearer explanation.*
LWP (Liquid Water Path) corresponds to the LWC integrated on the vertical. As LWC is overestimated near the ground (Fig.13) and as the fog layer is deeper, LWP is overestimated. It has been completed by : «  Due to the larger amount of cloud water near the ground, the dissipation at the ground is delayed by more than one hour . »

69. *Page 16, line 21: Is DE5 based on deposition on a grassy surface only, or is deposition over the entire tree canopy considered as well? In the context of this section, this text is not clear. Please clarify.*
DE5 was related to grass and tree canopy as it was like in REF. DE5 has been replaced by DE8 (deposition velocity of 8 cm/s) to answer to the newt point, and the principle has been clarified.

70. *Page 16, line 21: Why not use a value of 8 cm s-1, the upper bound suggested by Katata?*
OK, DE8 has been run and is presented instead of DE5. As explained in the introduction, the previous mistake on the deposition velocity has induced some modifications and now the DE8 simulation presents a significant improvement compared to REF.

71. *Page 16, line 22: Replace "diminution" by "reduction".*
OK.

72. *Page 16, line 29: Replace "the remove of" by "neglecting" for proper wording.*
OK.

73. *Page 17, line 26: I do not think "preformation" is a word. Maybe you mean "initial formation"?*
Thank you.

74. *Page 17, line 27: A DSD does not "move". Maybe "characterized by higher concentrations of larger droplets"*
Yes, thank you.

75. *Page 17, line 30: "dilutes" is not properly used here. You rather mean "decreases" or "diminishes".*
OK

76. *Page 17, line 30: Also this reduced effect impacts which field(s) in particular. Please*

*clarify.*
This has been clarified.

77. *Page 17, line 32, "fog slightly deeper": Please revise as "a slightly deeper fog layer"*
OK

78. *Page 18, lines 26-27: I do not understand the statement "diverged on the fog life cycle in the same way". Please revise your statement.*
This part has been removed as the text was too long.

79. *Page 18, lines 27-28: Not a very clear statement. Please revise. And be more explicit about what you mean by "dynamical conditions".*
This part has been removed as the text was too long.

80. *Page 19, line 10, "as the wind overcame this obstacle": Awkward formulation. Maybe "and associated perturbed mean flow and turbulence conditions" would be a clearer statement.*
OK, « overcame » has been replaced by « met »

81. *Page 19, line 17: replace "meeting" with "encountering" or "reaching".*
OK, "encountering".

82. *Page 19, line 17: use of the expression "dynamical gradients" is not specific enough. Do you mean "wind shear" in particular?*
Yes, thank you.

83. *Page 19, line 18, "became well-marked": This is awkward wording. Do you mean "became prominent"?*
Yes, this has been corrected.

84. *Page 19, line 23,"homogeneous". Where? Throughout the fog layer? At the top of the layer? Please be more precise in your statement.*
No, inside the cloud layer : « The cloud droplet concentration became quasi homogeneous in the fog layer when averaged over time but extremes of droplet concentration occurred locally near the top of the fog in the radiative cooling layer, with maxima preferentially upstream of the crests of the waves rather than downstream, in the ascent area. »

85. *Page 19, line 24: "evolved" rather than "involved"?*
Yes, thank you.

86. *Page 19, line 29: "damaging the visibility diagnostic" is awkward wording. Maybe "worsening visibility diagnostics"?*
Yes, thank you.

87. *Page 19, line 31: "The removal of the deposition process" is awkward wording. Maybe replace "The removal of" by "Neglecting".*
OK.

88. *Page 20, line 4, "Endly": You mean "Finally" or "Lastly"?*
Yes, lastly.

89. *Page 20, lines 4-5: The use of "reduce much more the number concentration" is awkward.*

*Change to "reduce the overestimated droplet number concentration" for a more precise statement.*
OK.

90. *Page 20, line 8: In what way "simulations remain very challenging"? Please explain.*
They are very challenging due to the importance to represent correctly surface heterogeneities. This has been corrected.

91. *Page 20, line 12: I suggest you replace "cannot be neglected anymore" by "should not be neglected"*
OK, thank you.

92. *Page 20, line 20: We should read "dewfall" instead of "dewfal"*
Yes, thank you.

93. *Page 20, line 23: Change "no one" to "none" for appropriate wording.*
OK

94. *Page 20, line 23: Change "to reproduce correctly" to "in correctly reproducing"*
OK.

95. *Page 39: The citation of Hammer is not accurate. That paper has now been fully published and the citation should now indicate : Atmos. Chem. Phys., 14, 10517- 10533, doi:10.5194/acp-14-10517-2014*
OK, thank you.

96. *The formatting of citations is inconsistent throughout the References section. In particular, the names of journals sometimes uses capital letters (as should be) and sometimes not. Please revise.*
Yes, it has been done.

*Note: Only the most important text corrections have been suggested. A much greater number of possible corrections have been omitted due to time constraints for the reviewer. I strongly recommend that the text be reviewed by someone with a higher proficiency in English.*

---

## Author Comment (AC2) · 31 Jan 2017

Dear Sir,

We would like to thank you sincerely for your precious support to correct the text, and all your suggestions. Before answering to your questions, we must confess that there was an error in the coding of the deposition process Âă: the deposition velocity was mistakenly multiplied by the volume of the grid, corresponding to a ratio of 25 for all the simulations at 5m resolution (so a deposition velocity of 50 cm/s instead of 2 cm/s was actually applied), and to a ratio of 4 for the simulation at 2m resolution (noted DX2). Consequently, the deposition effect was overestimated. All the simulations except the one without deposition (called NDG) have been run again and most of the figures have been updated. For the REF simulation (with a deposition velocity of 2 cm/s), the discrepancies with the observed microphysical fields are a bit stronger (cloud mixing ratio and droplet concentration more overestimated), but the DE8 simulation (deposition velocity of 8 cm/s as it was requested by one of the reviewers) presents a significant improvement. The signature of the fog onset at elevated levels in the REF simulation is not so marked, and is more evident in the DE8 simulation, showing that both the tree drag effect and the deposition are necessary to reproduce the formation of fog at elevated levels. The new results do not modify the analysis of the fog event and the conclusions of the study. The text has been also reduced to answer to the reviewersĂă: the sensitivity test on the initial conditions has been removed, as well as the corresponding figures. The length of the text has been reduced as expected. Lastly, the text has been revised by an english native speaker.

Best regards,

Christine Lac

Please also note the supplement to this comment:
http://www.atmos-chem-phys-discuss.net/acp-2016-900/acp-2016-900-AC2-supplement.pdf

**Supplement:**

We would like to thank you sincerely for your precious support to correct the text, and all your suggestions. Before answering to your questions, we must confess that there was an error in the coding of the deposition process : the deposition velocity was mistakenly multiplied by the volume of the grid, corresponding to a ratio of 25 for all the simulations at 5m resolution (so a deposition velocity of 50 cm/s instead of 2 cm/s was actually applied), and to a ratio of 4 for the simulation at 2m resolution (noted DX2). Consequently, the deposition effect was overestimated.
All the simulations except the one without deposition (called NDG) have been run again and most of the figures have been updated. For the REF simulation (with a deposition velocity of 2 cm/s), the discrepancies with the observed microphysical fields are a bit stronger (cloud mixing ratio and droplet concentration more overestimated), but the DE8 simulation (deposition velocity of 8 cm/s as it was requested by one of the reviewers) presents a significant improvement. The signature of the fog onset at elevated levels in the REF simulation is not so marked, and is more evident in the DE8 simulation, showing that both the tree drag effect and the deposition are necessary to reproduce the formation of fog at elevated levels. The new results do not modify the analysis of the fog event and the conclusions of the study.
The text has been also reduced to answer to the reviewers : the sensitivity test on the initial conditions has been removed, as well as the corresponding figures. The length of the text has been reduced as expected. Lastly, the text has been revised by an english native speaker.

*Recommendation: Major revisions required.*

*Overview: This manuscript presents an original and thorough examination of a fog event at a site with varying land use (grass and trees). The authors have used LES simulations to assess the impact of a line of trees on the formation and lifecycle of the fog. A variety of different simulations were used to determine which processes were having the largest effects on the fog, and this has resulted in improved understanding of this scenario, as well as some recommendations for improvements in further fog simulations.*

*The presentation of the manuscript could be significantly improved by being proofread by a fluent or native English speaker. There are many spelling and grammatical errors in the manuscript, as well as a considerable number of instances of awkward phrasing. The formatting of the manuscript is also inconsistent. These problems make the manuscript very hard work to read, and obscures the nuance and scientific value of the authors work, which is otherwise good.*

The text has been revised by a native speaker of English.

*Scientific comments:*
   1. *P3, line 33: What sort of profiler are you using?*
      This is a RPG-HATPRO water vapour and oxygen multi-channel microwave profiler : this information has been added.

   2. *P4, line 17: How do you differentiate between radiation fog forming under very low (150m) cloud, and cloud lowering to the surface? You describe this event as follows: "the cloud base height progressively subsided during about 30 min, until it reached the ground". This sounds indistinguishable from stratus fog.*
      You are right that the distinction between radiative fog and cloud lowering is not easy to make. Fog classifications traditionally use the Tardif and Rasmussen (2007) method. They differentiate stratus lowering from radiative fog by the wind speed and the cloud ceiling. If the wind speed at 10m is lower than 2.5 cm/s before the formation and the cloud ceiling is less than 100m then the fog is supposed to be radiative. Our measured wind speed at 10m is

under 2.5cm/s but our cloud ceiling is higher than 100m (150m). However according to Dupont et al. (2012), the lowering of a stratus is due to a cooling at its base by evaporation of sedimented droplets. Considering fall speed of 2.2 cm/s (Roach et al, 1976) it would necessitate at less 10 hours for the cloud to reach the ground. Moreover we believe that the cloud formation at 150m is due to the modification of the flow caused by the tree barrier resulting in an important vertical mixing on a significant depth. So we conclude that this fog is a radiative one.

We propose the text :

« As underlined by Stolaki et al. (2015), this characteristic is very common at Sirta and 88% of the radiation fog events during the field experiment were also elevated. However, they were not classified as stratus lowering as they were followed rapidly by formation of fog at the surface. A delay of 30 min between the formation at 150 m height and at the ground seems too short to be a stratus lowering, which is mainly driven by the evaporation of slowly falling droplets that cool the sub-cloud layer (Dupont et al., 2012). This suggests that this type of radiation fog could be linked with, and specific to, the configuration of the Sirta site. »

3. *P4, lines 24 & 25: It is not clear to which TKE measurement you are referring here. The increase in TKE at 10 m occurs 30 minutes before the increase at 30 m, not simultaneously. After this increase there is still quite a lot of variability in the TKE, so I would not describe it as constant.*
Yes, we agree. This has been corrected by :
« Around 0400 UTC, the TKE at 10 m height increased significantly, by 0.5 m²/s², and then presented some variability around this value, while maintaining a positive vertical gradient .»

4. *P5, lines 1 & 2: There is a 30 minute difference in timing between the increase in LWC and Nc.*
This 30 min difference was due to the minimum value of Nc used for the plot. In the revised paper, LWC has been replaced by the cloud mixing ratio, and the minimum values of rc and Nc plots have been reduced : there is no time lag anymore.

5. *P9, line 13: More detail about the temperature convergence is required – i.e. The temperatures measured at different heights converge.*
Yes, this has been corrected by : « At 0230 UTC, the apparition of fog at the ground was associated with a temperature homogenization in the first 30 metres, called temperature convergence by Price (2011) and corresponding to a neutral layer. »

6. *P9, line 14: If only RH is being considered, it is not accurate to say that fog formed at 0230, only that saturation was reached. You need to refer to e.g. a visibility measurement.*
Yes, we agree. A reference to the microphysical fields has been added.

7. *P9, line 22: This increase in TKE occurs > 30 minutes before the TKE increase in the observations.*
Yes, you are right, this advance of 30 min corresponds to the advance of 30 min on the formation of fog near the ground : the remark has been added : « Around 0300 UTC, a more sudden increase of TKE occurs, as in the observations but 30 min before and with a lower magnitude. »

8. *P10, line 25: Please define the difference between sedimentation and deposition.*
Sedimentation corresponds to the gravitational settlement of droplets (it has been added in 2.3.1), while deposition represents direct droplet interception by the plant canopies (already defined in 2.3.1).

9. *P11, line 9 and onwards: You keep switching between LWC and rc throughout the manuscript. It would be better to consistently use one or the other.*
Yes, we agree. Only cloud mixing ratio is now used throughout the paper.

10. *There are a few statements throughout the manuscript which are accompanied by "not shown". Is there a particular reason why they are mentioned, but not included in plots?*
No there is no particular reason, but only to limit the number of figures. The number of « not shown » has been reduced.

11. *P15, line 6: What do you mean by the "production" of Nc?*
« Production » of Nc corresponds to the positive temporal evolution of Nc, considering the prognostic evolution of this field. Production terms of Nc come from activation, accretion, autoconversion, evaporation and sedimentation as presented in Khairoutdinov and Kogan (2000) and Geoffroy et al. (2008). A mention has been added as : *« Inside the fog layer, despite the increase of $r_c$ , the positive temporal evolution of $N_c$, called the production of $N_c$ is not higher than in REF »*

12. *P16, line 5: In the context of fog microphysics, 3m is not especially "near surface".*
Yes, this has been corrected.

*Technical comments:*
*1. Section 2.3.1: Not all terms of the equations presented in this section are defined in the text.*
Thank you, definition of S and $\rho_a$ have been added.

*2. Section 2.2.3: The figure numbers in this section do not correspond to any of the figure captions.*
We suppose you speak of Section 2.3.3 and figure A.2 is given in the Appendix on Material support.

*3. P9, lines 17-19: It is difficult to see the negative temperature gradients in Fig. 2, due to the number of lines.*
The number of lines has been reduced with only 1m, 5m and 30m.

*4. P9, lines 20 & 22: Please refer to Fig. 3a & 3b, instead of just Fig. 3.*
OK.

*5. P11, lines 15-17: It would be helpful to the reader if the different phases of the fog lifecycle were marked on any plots showing a time series of data.*
OK, the 3 phases have been plotted on the (z,t) plots.

*6. P11, line 30: "when the fog reached approximately 80 m". Is this the depth of the fog, the height of the fog top, or the location of the cloud/fog base?*
Yes, this is the depth of the fog and also the height of the fog top. It has been corrected.

*7. P11, line 33: Fig. 7d shows updraft velocity, not cooling.*
Yes, thank you.

*8. Section 3.3: Marking the location of the trees on plots of spatially varying data would make the figures easier to interpret.*
OK, this has been added on Fig.9 and 12.

*9. Please put the figures in the order in which they are first referred to in the text.*
This has been corrected.

*10. There are numerous occasions where the figures are incorrectly referenced in the text. Please correct this.*
This has been corrected.

*11. When plotting a time series from the LES, please state where in the domain the data was from.*
This has been added.

*12. P18, line 12: Are you referring here to the surface, or 3m?*
You are right, it is 3m.

*13. References: The capitalisation of journal titles and place names in the reference list is inconsistent, there are also some references missing page numbers.*
Yes, it was a problem of Latex and it has been corrected.

---

## Author Comment (AC3) · 31 Jan 2017

Dear Sir,

We would like to thank you sincerely for your precious support to correct the text, and all your suggestions. Before answering to your questions, we must confess that there was an error in the coding of the deposition process : the deposition velocity was mistakenly multiplied by the volume of the grid, corresponding to a ratio of 25 for all the simulations at 5m resolution (so a deposition velocity of 50 cm/s instead of 2 cm/s was actually applied), and to a ratio of 4 for the simulation at 2m resolution (noted DX2). Consequently, the deposition effect was overestimated. All the simulations except the one without deposition (called NDG) have been run again and most of the figures have been updated. For the REF simulation (with a deposition velocity of 2 cm/s), the dis-

crepancies with the observed microphysical fields are a bit stronger (cloud mixing ratio and droplet concentration more overestimated), but the DE8 simulation (deposition velocity of 8 cm/s as it was requested by one of the reviewers) presents a significant improvement. The signature of the fog onset at elevated levels in the REF simulation is not so marked, and is more evident in the DE8 simulation, showing that both the tree drag effect and the deposition are necessary to reproduce the formation of fog at elevated levels. The new results do not modify the analysis of the fog event and the conclusions of the study. The text has been also reduced to answer to the reviewers : the sensitivity test on the initial conditions has been removed, as well as the corresponding figures. The length of the text has been reduced as expected. Lastly, the text has been revised by an english native speaker.

Best regards,

Christine Lac

Please also note the supplement to this comment:
http://www.atmos-chem-phys-discuss.net/acp-2016-900/acp-2016-900-AC3-supplement.pdf

**Supplement:**

We would like to thank you sincerely for your precious support to correct the text, and all your suggestions. Before answering to your questions, we must confess that there was an error in the coding of the deposition process : the deposition velocity was mistakenly multiplied by the volume of the grid, corresponding to a ratio of 25 for all the simulations at 5m resolution (so a deposition velocity of 50 cm/s instead of 2 cm/s was actually applied), and to a ratio of 4 for the simulation at 2m resolution (noted DX2). Consequently, the deposition effect was overestimated.
All the simulations except the one without deposition (called NDG) have been run again and most of the figures have been updated. For the REF simulation (with a deposition velocity of 2 cm/s), the discrepancies with the observed microphysical fields are a bit stronger (cloud mixing ratio and droplet concentration more overestimated), but the DE8 simulation (deposition velocity of 8 cm/s as it was requested by one of the reviewers) presents a significant improvement. The signature of the fog onset at elevated levels in the REF simulation is not so marked, and is more evident in the DE8 simulation, showing that both the tree drag effect and the deposition are necessary to reproduce the formation of fog at elevated levels. The new results do not modify the analysis of the fog event and the conclusions of the study.
The text has been also reduced to answer to the reviewers : the sensitivity test on the initial conditions has been removed, as well as the corresponding figures. The length of the text has been reduced as expected. Lastly, the text has been revised by an english native speaker.

*This paper presents large eddy simulations of a radiation fog event for which extensive research quality observations were available. The main focus of the paper is to uncover how different aspects of the model dynamics affect the fog evolution, and sensitivity to the surface treatment, initial conditions and model dynamical formulation are investigated. Whilst the work is interesting, and ultimately worthy of publication, I feel extensive modifications to the manuscript are required before it is suitable for publication.*

*Firstly, the manuscript is very difficult to read, due to numerous spelling and grammatical mistakes. A revised version would benefit from extensive proof-reading and typographical editing, possibly with the help of a native English speaker.*

The text has been revised by a native speaker of English.

*I have provided suggestions for the abstract below, to give the authors an idea on the level of modification required:*

*L2 - should say "...during the ParisFog..."*
*L4 - should say "...of a tree barrier..."*
*L7 - should say "...as in the observations, and..."*
*L10 - should say "...meaning that grid convergence..."*
*L12 - should say "...and had a similar effect to removing the tree barrier."*
*L13 - should say "...allows us to..."*
*L13 - should say "...necessary to correctly simulate the fog life cycle at high resolution."*

OK, thank you.

*Secondly, the manuscript lacks structure and coherence. It currently just presents a long list of things you have done, with no real theme linking everything together or justifying the various experiments. The introduction should focus on the specific problem you are trying to address - how dynamics affects the evolution of fog, what specific questions are you trying to answer? This should then provide justification for the sensitivity experiments you conduct - how do they help you answer the questions?*

*The conclusions should then tie all this together and answer those questions. It is possible that in doing this, you may be able to shorten the text (which is currently quite long) and number of figures, to only focus on what is really relevant.*

In the introduction, this sentence has been introduced : « In order to establish the main ingredients driving the fog life cycle and the microphysical fields, and to evaluate how dynamics affects the evolution of fog, sensitivity experiments are conducted with the model considered as a laboratory. »

In the conclusion : «  Various sensitivity tests allowed to identify the main processes affecting the evolution of fog. »

The text has also be shortened as the sensitivity tests on the initial fields have been removed. It should give more structure and coherence to the paper.

*I only have two specific scientific comments:*
*Sect 2.3.2 - why do you choose an empirical diagnosis of visibility based on the cloud water content and drop number, rather than calculating the visibility accurately from Eqn. 7? With the complicated microphysics scheme you have available, you should be able to calculate the extinction coefficient directly, e.g. as done by Clark et al. (2008).*

You are right but the objective here was not to calculate the visibility as accurately as possible but to estimate the best diagnostic relationship often used by the models.

*P9, L32 - do you have observations of the surface or soil temperatures which you could compare to the model here to explain the difference in upwelling LW radiation?*

We have observations of the surface and  soil temperatures but we do not give them a good degree of confidence as they present strong differences with 1m temperatures.

---

## Author Comment (AC4) · 31 Jan 2017

[revised manuscript text omitted]
 was reduced by 2%, and, in the second, called HP3, the relative humidity of the initial profile was increased by 3% over the same depth. In Fig. 13g, h and i, it appears that the fog life cycle is significantly modified, with a fog onset time deviating from the observations : it occurs around 2 hours earlier with HP3 and 2 hours later with HM2. However 3 m $r_c$ is almost the same during the development and mature phases. Also neither of the simulations changes the DSD or the droplet concentration extrema. The LWPs of REF, HM2 and HP3 are superimposed (Fig. 15c) during the mature phase, so the dissipation time is unchanged.~~

~~The last test involved an increase (VP3) or a decrease (VM3) of the wind speed in the free atmosphere in the initial and forcing conditions. In Fig. 13g, h and i, it can be seen that the lower the wind, the earlier the formation time, the higher the $r_c$ and the later the dissipation time, as the mixing with higher dry, warm air is reduced. In contrast, a stronger wind drastically reduces the duration of the fog life cycle and the surface $r_c$. VM3 succeeds in broadening the droplet spectrum, but the extrema of the droplet concentration do not change significantly.~~

~~Thus, all the tests presented in Figures 13 and 15 fail to reduce the droplet concentration compared to REF. Only the NTR simulation reduces it somewhat, due to a broader droplet spectrum, but it overestimates the $r_c$ and advances the fog onset. This probably means that modifying the dynamical conditions is not a way to improve the droplet concentration prediction further, 
[revised manuscript text omitted]

---

## Author Response (AR2)

**Answer to Co-Editor and Reviewers : acp-2016-900**

**We thank the Co-Editor and the three reviewers for their comments. We answered below to all their points. Their comments are in italic font while our answers appear in blue normal font. Changes made to the original version of the paper appear in track-change mode on the enclosed pdf.**

*Co-Editor Decision: Reconsider after major revisions (15 Jun 2017) by Patrick Chuang*

*Comments to the Author:*
*Major revisions required, including editing for language and a response to editor comments. See the attached file for a partially-edited manuscript showing examples of language edits.*

*Non-public comments to the Author:*
*While the reviewers seem inclined to accept the manuscript, one of them points out that there are English problems that still need to be resolved. I started editing the manuscript in response to this point and, as you will see, I got through the first 6 pages, in which I made nearly 100 edits, which took me more than 3 hours. There are some grammatical and spelling errors, but many more involve awkward, imprecise, or unclear wording. Some of this issue is, I'm sure, due to English not being a first language for the authors, for which I am sympathetic. Some of it, however, is not because of this, such as the many careless errors, and I think there is a serious lack of attention to detail that is inappropriate for a scientific manuscript submission.*

*You need to have someone actually properly edit the remainder of the manuscript to make sure that the English is of high quality. I know you said that an English speaker did make edits to the original manuscript, and while it clearly was improved, that person did not do a sufficiently good job. You may need to find someone more qualified to help you. ACP does have a (paid) language service, but I have not used it so I not sure what it can and can't do for you. In any case, you need to find some solution for this issue.*

*I was initially hoping I could do the edits myself, but after 3 hrs of my time, I felt that I had spent more time than is reasonable helping you make them.*

Thank you very much for the time and the effort you have put into the correction of this paper. The English of the manuscript has been corrected a second time by an English speaker, we hope it would be correct now.

*A second issue is that the gamma distribution used to represent the drop size distribution (DSD) does an extremely poor job. Yet one of the main goals of the study is to understand the impact of dynamics on the microphysics. If the simulated DSD is completely different from the observed, then how can you possibly properly simulate, e.g. the radiative budget or the deposition, both of which are sensitive to the DSD, and also will feedback to the dynamics? This seems like a potentially serious weakness of your study.*

We agree with you that the improvement of the simulated microphysical fields is a challenge. As explained in the introduction, most of the studies fail in reproducing realistic liquid water contents and droplet concentrations near the ground. When liquid water contents and droplet concentrations are overestimated, the DSD cannot be realistic. That is why we focused on the impact of dynamics, deposition, and numerical schemes on the microphysics. And this paper shows that these ingredients

are necessary to be well treated, and can improve the microphysicals fields. But they are not sufficient. As shown by Thouron et al. (2012) for stratocumuli, a prognostic treatment of the supersaturation would be necessary to improve more significantly the microphysicals fields : this will be presented in a second part.
However, even if the simulated DSD is completely different from the observed one, Fig.4 shows that the model reproduces fairly well the radiative fluxes. Additionally, there is no upper air microphysical measurement, but a possibility is that simulated DSD could be better above than near the ground.

*Other comments:*

*The figures need some improvements before they are publication quality. In many cases, the font of the legend (e.g. Fig 2) or the axes (e.g. Fig 2 and 5) are too hard to read. Some combination of font size and line thickness should fix this problem. Also, I don't think it's acceptable to have the units lacking the the proper superscripts (e.g. m2/s2 not having the exponents in the correct position). This occurs in many of the plots.*

All these aspects have been corrected.

*In some places, your units are in italics, and in other places they are in regular font. The latter is correct, while the former is not. Please fix all occurrences. This is one example of the lack of attention to detail that is unrelated to English as a second language.*

This has been corrected.

*As in my comments, in some places you have references like (Lac (2013)), while others it would read (Lac, 2013). Please make them all consistent, preferably following the latter example. This is another example of the lack of attention to detail that is unrelated to English as a second language.*

This has been corrected.

*You say that the lateral boundary conditions are cyclic. Does that mean that the simulations include trees on both sides, while in reality they appear only on one side (at least that's how it looks in Fig. 1)? If so, what are the consequences of this incongruence?*

Yes, the lateral boundary conditions are cyclic and there is no incongruence, as the surface conditions are the same at both sides of X, and both sides of Y : trees are present between X=100m and X=200m for Y=0m and Y=1000m, and there is no tree at X=0m and X=1000m.

*You refer to equations inconsistently. It should be "Eq. 2", not "Eq.2" or "equation (2)". Please fix all occurrences.*

This has been corrected.

*Additional remarks from the pdf file reported here:*

*Abstract : Do these heterogeneities disappear after formation?*

They are mainly marked at the formation. Then, during the development phase, some differences between the two simulations, with or without trees, appear on the Kelvin-Helmholtz waves, but they do not impact strongly the fog development. So we cannot say that these heterogeneities completely

disappear.

*Part 2.1 : How is it possible to measure upward fluxes from a roof?*

You are right : only the downward fluxes were measured from the roof, and upward fluxes were measured from a mast at the same location. The reference to the roof has been removed.

*Part 2.1 :Is the the visibility really measured in the vertical direction? That seems extremely unlikely.*

You are right, « vertical » has been replaced by « horizontal » in the text.
More specifically, a Degreanne DF20+ diffusometer is operated near the ground (3 m agl) and a second  Degreanne DF20 diffusometer is operated at 18 m agl, providing measurement of visibility with a +/- 10-25 % uncertainty (Stolaki et al., 2015).

*Part 2.2.2 : As above, I don't see a second mode. I'm very skeptical of calling these distributions "bimodal".*
It is indeed a possibility, we now speak of a broadening of the distribution, and not a second mode.

**Reviewer #1**

*I commend the authors on their revisions to the manuscript, it is now much improved. It should be suitable for publication, subject to some minor revisions:*

*- the authors reference Mazoyer et al (2016) throughout the manuscript. Given this paper was rejected for publication in ACP, I don't really feel it should be referenced at all. But if it absolutely must be referenced, it should be done so as "unpublished manuscript" or made clear that it has failed the peer-review process - at present it is given similar weight to all other published references.*
Mazoyer et al. (2016) is not rejected for publication in ACP, but you are right that there has been a delay in the revision: a revision of this discussion paper for further review has not still been submitted. It is reference as ACPD, and we would prefere to not include the unpublished mention.

*- Eq. 7: I accept why the authors do not wish to diagnose visibility from this equation, but feel they should at least reference Clark et al. (2008, doi:10.1002/qj.318) to demonstrate that this method can (and is) used to predict visibility correctly in NWP models, and we don't need to rely on the crude empirical approximations of equations 8 and 9.*
This reference has been included as : « Clark et al. (2008) use this equation to predict visibility correctly in an NWP model. »

*- Section 4.4 seems to be included with no content...*
It was just to show that this part had been deleted from the previous version, following recommendations of the Reviewers.

**Reviewer #2**

*Although the English is much improved, there are still a lot of errors. I suggest that the authors get the manuscript proofread professionally.*

The English of the manuscript has been corrected a second time by an English speaker, we hope it would be correct now.

*There is a feature described as "noteworthy" in section 3.2, but it is noted that this is not shown. If the feature really is noteworthy, a figure should be included.*

It was a previous recommendation to limit the number of figures, so we have not added a new figure. But we have removed the « noteworthy » term.

[revised manuscript text omitted]

---

## Author Response (AR3)

**Answer to Co-Editor and Reviewers : acp-2016-900**

**We thank the Co-Editor for his comments. We answered below to all the points. The comments are in italic font while our answers appear in blue normal font. Changes made to the original version of the paper appear in track-change mode on the enclosed pdf.**

*Co-Editor Decision: Reconsider after major revisions (04 Aug 2017) by Patrick Chuang*

*Comments to the Author (pdf): acp-2016-900-comments-to-author.pdf*

*Comments to the Author:*

*See attached marked-up manuscript.*

*Non-public comments to the Author:*

*The English is much improved. I've flagged a few more issues (see attached marked up manuscript), but there are certainly a lot fewer than before. I would suggest to the senior authors that in the future, they need to ensure that the quality of the writing is at this level \*before\* the initial submission. The poor writing quality has made things much harder for everybody involved -- authors, reviewers and the handling editor.*

Thank you again for the time and the effort you have put into the correction of the English of the paper.

*My remaining concern is the poor simulation of the microphysics. You state that the simulations are off by a factor of 14 in Nc, and peak LWC is off by a factor of 7. This is not "acceptable" as you write. I would call it unacceptable, especially when I likely could have blindly guessed the numbers to within a factor of 2. The disagreement between simulated and observed drop size distributions further emphasizes the lack of confidence that many readers would have in the simulated microphysics.*

*Given that, I request that you do the following:*

*(a) Spend more time justifying why your simulation results should be considered reasonable given the disagreement in microphysical properties AND*

*(b) Remove any substantial discussion and figures of aspects of the simulation that are sensitive to the microphysical properties. To delve deeply into Nc, rc, visibility, supersaturation, sedimentation or deposition rate, etc. is, I think, a poor strategy given the simulation-observation mismatch. This includes aspects like the sedimentation sensitivity study, or the section on the effect of deposition. There's plenty of material in the manuscript as it is that a shift in the focus will still make for an interesting, and stronger, manuscript (in my opinion).*

*(c) Keep the parts of the manuscript that discuss results that are better simulated, like the dynamics, entrainment, surface fluxes, etc. There are certainly feedbacks between microphysics and dynamics/thermodynamics, but I still think you can make a good case (as in (a)) that they aren't so important that your results are invalid.*

*At this point, these are not suggestions. Since you are planning a Part 2 for this manuscript,*

*perhaps you can transfer to that manuscript anything you wish to say that I'm not letting you discuss here.*

*If you disagree with my view and want to push back, I'm happy to have a short conversation by email (feel free to email me directly at pchuang@ucsc.edu). But it's pretty unlikely that I will change my mind on this. I've spent many years working in the space between cloud microphysical observations and LES results (albeit not specifically looking at fog), and I would never consider analyzing these simulations given their poor agreement with observations. But since that's where we are, and I assume you are not interested in generating new and improved simulations, I think this is the best we can do working with what you have.*

*By the way, I won't be sending the manuscript out for review for the next round. The next review will only be mine, which should speed things up.*

We agree with you that the agreement between the observed and the simulated microphysical fields near the ground is very poor. So we have taken your recommendations. We have removed the simulation-observation comparison of the near surface microphysical fields and we have kept the LWP comparison as it is representative of the whole fog depth. Our reference simulation gives good agreement with near-surface wind, temperature, humidity and LWP, so we can be confident in this simulation. All the sensitivity tests can be evaluated with the LWP measurements and the main conclusions are kept unchanged.

We also propose to keep the vertical description of rc, Nc and supersaturation as the LWP comparison shows a good behaviour of the model, apart from the near-surface microphysics. Even if, contrary to rc, there is no altitude or vertically integrated measurement of Nc, the fact that the total amount of water is correct leads us to be confident in the vertical and temporal variations of Nc (even if doubts may remain on the absolute values). So we can consider that the vertical evolution of the microphysics is realistic.

It means that one figure has been removed and another one has been reduced. The text is also reduced and easier to read. We have also the feeling that the paper has a greater quality of purpose : the objective is not to improve the near-surface microphysical fields any more, but to characterize the most important dynamical ingredients characterizing the radiative fog life cycle. Consequently, we propose to change the title of the paper : « Impact of dynamics on the fog life cycle » will replace « Impact of dynamics on the microphysics » and the naming « Part 1 » has been deleted.

By the way, we keep the idea to explore more the near-surface microphysics in a further study in order to understand why the model is not able to reproduce the low measurements values of rc and Nc. Theses values may be determined by very local features near the ground. As the near-surface gradients of rc and nc are large (see fig 7 and 9), large errors at the height of measurement can be induced by not so large surface processes (or a combination of these). Some areas for improvement are mentionned in the last part of the conclusion : fog water deposition could be strongly enhanced at the forest edge, and could consider the hygroscopic nature of canopies.

Additionally, surface heterogeneities are very complex at the Sirta site : trees but also small buildings. Lastly, a multimodal approach of aerosols (Vié et al., 2016) may be of benefit to better simulate the fog microphysics and the multimodal Particle Size Distribution, but it is out of scope of this paper.

We have the feeling that the paper is improved so we would like to thank you sincerely for the recommendations.

[revised manuscript text omitted]

---

## Author Response (AR4)

**Answer to Co-Editor and Reviewers : acp-2016-900**

**We thank the Co-Editor for his comments. We answered below to all the points. The comments are in italic font while our answers appear in blue normal font. Changes made to the original version of the paper appear in track-change mode on the enclosed pdf.**

*Comments to the Author:*
*Thank you for making those revisions. I think the text is now in acceptable shape for the most part.*

*Here are my (hopefully) last round of requested edits:*

*\*\*\* Most importantly, the font (words or numbers of both) on your figures is still too small in many places. Figs 1, 2, 9, 10, 13, A.1, A.2 all clearly need to be fixed, though almost all your color map axis/colorscale values probably need to be larger. I've requested this before and you didn't really fix the problem. Please fix all of these to be suitably readable. \*\*\**

**This has been corrected.**

*In addition:*

*1. Please state clearly that simulated drop concentrations are much larger than observed (possibly at the top of p. 10). This is an important caveat, and also something that readers would wonder if they examine Fig. 6b.*

**A sentence has been added at the beginning of 3.2: « The vertical and temporal variations of simulated Nc can be studied as the LWP is realistic, but values of Nc must be carefully considered as a first comparison to near-surface measurements clearly shows an overestimation of simulated values. »**

*2. The lines, colors and dashes are not consistent in Figs. 2 to 5. For example, you have 3 sets of curves in Fig. 2 with colors black, purple and green. In Fig. 3 your 3 colors are black, red and blue. In Fig 2, obs vs model are solid vs dashed while colors are for different altitudes. But in Fig 3 this is opposite, with obs vs model are different colors, while solid vs dashed is for different altitudes. Make your curves consistent across all these line figures.*

**This has been corrected.**

*3. Fig. 9c seems to have a different color bar than all the other colormap figures. Make it the same.*

**This has been corrected.**

*4. Go through your references and carefully edit it for errors. For example the Porson and Price citations both have "i" or "ii" where it should be "I" or "II". Lafore citation uses "Vol" where it should be boldface.*

**This has been revised.**

*5. All your references use the full name of the journal. Change every one to the proper abbreviated version, so ACP = Atmos. Chem. Phys. instead of fully spelled out.*

**This has been revised.**

*6. On Figures 2 to 5 and Fig. 12, please add error bars for the observational curves. It may be easier to do a lightly shaded area rather than bars for every point since your data has such a high time resolution.*

**Grey areas have been added to the observational curves.**

Thank you again for the time and the effort you have put into the correction of the paper.